# Energy and Memory-Efficient Federated Learning with Ordered Layer Freezing and Tensor Operation Approximation

## Abstract

The effectiveness of Federated Learning (FL) in the context of the Internet of Things (IoT) is hindered by the resource constraints of IoT devices, such as limited computing capability, memory space and bandwidth support. These constraints create significant computation and communication bottlenecks for training and transmitting deep neural networks. Various FL frameworks have been proposed to reduce computation and communication overheads through dropout or layer freezing. However, these approaches often sacrifice accuracy or neglect memory constraints. In this work, we introduce Federated Learning with Ordered Layer Freezing (FedOLF) to improve energy efficiency and reduce memory footprint while maintaining accuracy. Additionally, we employ the Tensor Operation Approximation technique to reduce the communication (and accordingly energy) cost, which can better preserve accuracy compared to traditional quantization methods. Experimental results demonstrate that FedOLF achieves higher accuracy and energy efficiency as well as lower memory footprint across EMNIST, CIFAR-10, CIFAR-100, and CINIC-10 benchmarks compared to existing methods.

## 1 Introduction

Federated Learning (FL) (McMahan et al., 2017) has gained significant traction in the Internet of Things (IoT) for processing decentralized data and providing privacy-preserving intelligent services to clients (Jin et al., 2024; Zheng et al., 2023; Nguyen et al., 2022). However, the heterogeneous nature of client devices poses a challenge due to varying system capacities. In real-world IoT environments, clients, often edge devices, exhibit diverse configurations in terms of processor, battery, bandwidth, and memory. Resource-constrained devices with limited hardware and bandwidth face difficulties in training and transmitting large neural networks, leading to straggling, low quality-of-service, and excessive computation and communication costs. Moreover, devices with insufficient memory may be unable to handle memory-intensive neural networks, thus being excluded from FL with severe information loss. Therefore, addressing the issue of resource constraints is crucial for the successful application of FL in IoT systems (Imteaj et al., 2022; Pfeiffer et al., 2023a).

Several studies have been proposed to address resource constraints through techniques such as **dropout** (Caldas et al., 2018; Horváth et al., 2021; Diao et al., 2021; Kim et al., 2023) or **layer freezing** (Pfeiffer et al., 2023a;b). These methods involve training a subset of the global model with reduced requirements on hardware, bandwidth, and memory on edge devices. Specifically, dropout involves pruning a fraction of the global model and sending the remaining sub-model to clients for training. However, it may significantly degrade accuracy in non-independent identical (non-iid) local data distributions. In such settings, data importance among clients may vary, and training an underparameterized sub-model for an important client with data resembling the global distribution may not sufficiently capture knowledge from local data, leading to decreased accuracy of the global model (Pfeiffer et al., 2023b; Acar et al., 2021).

Instead of sub-models, layer freezing involves sending the full global model to all devices and allowing resource-constrained devices to freeze some layers during training. For example, CoCoFL (Pfeiffer et al., 2023b) allows clients to randomly train certain layers while freezing the remaining, while SLT (Pfeiffer et al., 2023a) enables clients to sequentially train each layer in a bottom-up man-

ner, with other layers partially frozen. Compared to dropout, layer freezing is more resilient to non-iid data by preserving the full model architecture on each client (Pfeiffer et al., 2023b). However, layer freezing introduces heavy communication overhead since the global model must be transmitted to clients. Additionally, these methods overlook the fact that top-level layers, even though frozen, still need to store and pass gradient information back to lower-level active layers during backpropagation, resulting in heavy memory consumption. For example, Figure 1 illustrates a comparison between two training modules: (a) random layer freezing requiring more memory than (b) ordered layer freezing due to a longer path for backpropagation of gradients (a longer red arrow in Figure 1(a)). To validate this analysis, we implement these two layer-freezing strategies using ResNet20 (He et al., 2016) with the CIFAR-100 dataset (Krizhevsky et al., 2009), and measure their maximum memory usage using the TORCH.CUDA.MAX_MEMORY_ALLOCATED function (PyTorch, 2023) in PyTorch. As depicted in Figure 1(c), random layer freezing consumes more memory compared to ordered layer freezing, even when the same number of layers are frozen.

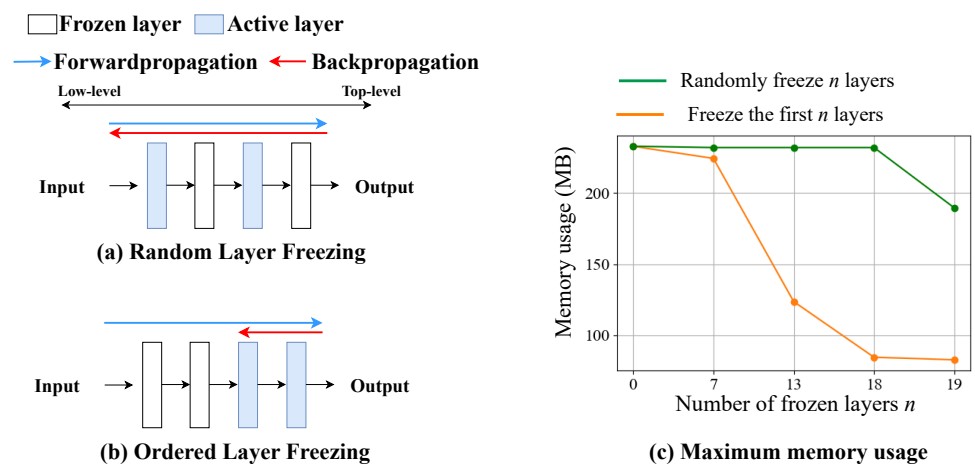

Figure 1: A comparison between **(a)** Random Layer Freezing and **(b)** Ordered Layer Freezing. The former requires more memory space to pass the gradient information back towards low-level active layers. **(c)** shows the maximum memory usage of these two modules in practice.

To address the shortcomings of existing methods, we introduce a new FL framework named **Federated Learning with Ordered Layer Freezing (FedOLF)**. In FedOLF, resource-constrained devices selectively freeze some low-level layers while training the remaining top-level layers. This approach substantially reduces the computation overhead and memory requirements of training, by shortening the gradient backpropagation path as illustrated in Figure 1(b). Additionally, we empirically observe that the gradient loss resulting from low-level frozen layers tends to diminish as training moves forward to top-level layers, which helps FedOLF maintain accuracy. Furthermore, we adopt an adapted **Tensor Operation Approximation (TOA)** scheme (Adelman et al., 2021) to reduce the communication cost in FedOLF. Instead of the full global model, clients receive a low-rank approximation of the frozen layers along with all active layers from the server during communication. Unlike conventional quantization methods, TOA minimally impacts training and significantly preserves model accuracy. The contributions of this paper are summarized as follows:

- We introduce FedOLF, an efficient FL framework addressing the memory shortage problem by allowing resource-constrained devices to train partial top-level layers of the global model. We also provide convergence analysis of FedOLF in non-convex settings.

- We propose an adapted TOA framework to reduce communication costs and memory footprint of FedOLF. Unlike the initial method that works on all layers, the adjusted TOA framework only works on frozen layers to ensure the active layers get fully trained.

- We evaluate FedOLF on EMNIST (with CNN), CIFAR-10 (with AlexNet), CIFAR-100 and CINIC-10 (with ResNet20 and ResNet44). Experimental results demonstrate that FedOLF outperforms the state-of-the-art by improving the accuracy by at least 0.3%, 6.4%, 12.8%, 4.4%, 6.6% and 1.29%, with higher energy efficiency and lower memory footprint.

## 2    Literature Review

**Efficient Federated Learning:** This stream of research aims to alleviate the computational and communication costs associated with FL. Various approaches have been proposed to enhance computation efficiency, such as FedProx (Li et al., 2020), FedParl (Imteaj & Amini, 2021), and PyramidFL (Li et al., 2022), which reduce client training epochs to mitigate computation costs. To improve communication efficiency, methods like FedCOM (Haddadpour et al., 2021), FetchSGD (Rothchild et al., 2020), and STC (Sattler et al., 2020) reduce the size of transmitted parameters through message compression. Additionally, approaches like FedSL (Zhang et al., 2024), FedOBD (Chen et al., 2022a), FedNew (Elgabli et al., 2022), Fedproto (Tan et al., 2022), and DS-FL (Itahara et al., 2023) advocate for transmitting lightweight replacement messages, such as logits and prototypes, instead of the full global model. However, these methods often focus on singular aspects of efficiency and fail to simultaneously address both computation and communication challenges. Moreover, they do not adequately account for memory constraints on devices, as they typically involve full-model training on all clients. Adaptive dropout (Li et al., 2021a; Jiang et al., 2022; 2023; Li et al., 2021b) offers a more comprehensive approach by enabling clients to train and transmit lightweight sub-models, thereby achieving both computation and communication efficiency. Nevertheless, adaptive dropout overlooks memory constraints, as clients must prune unimportant neurons to generate sub-models, a process that requires pre-training the full model locally. FLrce (Niu et al., 2024) mitigates overall computation and communication costs by reducing FL iterations with an early-stopping mechanism. However, it still entails full-model training on all devices irrespective of memory constraints.

**Federated Learning on Resource-Constrained Devices:**    The primary distinction between efficient FL and resource-constrained FL lies in the latter's consideration of devices with limited resources, such as memory space or bandwidth support, which are unable to train or transmit the entire model. To tackle this challenge, (Caldas et al., 2018; Horváth et al., 2021; Kim et al., 2023; Diao et al., 2021) introduce the concept of sub-models, which contain fewer parameters and can be trained and transmitted by resource-constrained clients. Specifically, Feddrop (Caldas et al., 2018) employs random neuron pruning, FjORD (Horváth et al., 2021) and HeteroFL (Diao et al., 2021) adopt a right-to-left approach for neuron pruning, and DepthFL (Kim et al., 2023) employs top-first layer pruning. Unlike adaptive dropout, these works execute dropout at the server side, eliminating the need for clients to pre-train a full model. However, these methods are susceptible to non-iid data among clients, as training small sub-models on crucial clients may not capture sufficient knowledge to construct an accurate global model. In contrast, CoCoFL (Pfeiffer et al., 2023b) and SLT (Pfeiffer et al., 2023a) advocate for maintaining the full model architecture on all clients while freezing certain layers on resource-constrained devices. CoCoFL randomly freezes layers within the local model, whereas SLT partially freezes top-level layers and sequentially trains all layers from the bottom. The frozen layers remain untrained and untransmitted to enhance computation and communication efficiency. However, these approaches lead to increased memory usage, particularly in the case of frozen top-level layers, which consume significant memory space to transmit gradient information backward, as illustrated in Figure 1.

## 3    Methodology

### 3.1    Problem Setup

Given a network with one server and $K$ devices (clients), and a global model $w$ stored on the server side, the goal of FL is to optimize the following problem:

$$\min_w f(w) := \mathbb{E}[f_k(w)] := \sum_{k=1}^{K} \frac{n_k}{n} (f_k(w)),$$

$$f_k(w) := \frac{1}{n_k} \sum_{n_k}^{i=1} \mathcal{L}(w, (\boldsymbol{x}_i, y_i)). \tag{1}$$

$f$, the global objective function, is a weighted average of all local objective functions $f_k$ ($1 \leq k \leq K$). For a client $k$, the local objective function $f_k$ is equivalent to the empirical risk over its personal dataset $D_k$, $n_k = |D_k|$ is the size of the local dataset and $\mathcal{L}(w, (\boldsymbol{x}_i, y_i))$ is the prediction loss of

$w$ over the $i-$th sample $(\boldsymbol{x}_i, y_i)$ in $D_k$. $n = \sum_{k=1}^{K} n_k$ is the total number of samples across all local datasets. Moreover, let $N$ denote the total number of layers in the global model $w$, and $W_l$ represent the $l-$th layer with parameter $\boldsymbol{\theta}_l$ $(1 \le l \le N)$. The layer $W_l$ can be viewed as a function that takes the input feature representation $\boldsymbol{x}_{l-1}$ from the previous layer, and outputs a new feature representation $\boldsymbol{x}_l$, i.e. $\boldsymbol{x}_l = W_l(\boldsymbol{x}_{l-1}, \boldsymbol{\theta}_l)$. Specially, $\boldsymbol{x}_0 = \boldsymbol{x}$ is the initial data sample, and $\boldsymbol{x}_N = \hat{\boldsymbol{y}}$ is the model's final prediction.

---

**Algorithm 1** FedOLF

---

**Require:** maximum global iteration $T$, clients $C = \{1, ..., K\}$ with numbers of frozen layers $\{l_1, ..., l_K\}$, and initial global model $w^0$, scale factor $s$.

1: **for** $t = 1, 2, ..., T$ **do**
2:     Server randomly samples a set of participating clients $C_t \subset C$.
3:     **for** every client $k \in C_t$ **the server does**:
4:         Decompose $w^t$ into $w_{F,k}^t$ and $w_{A,k}^t$ based on $l_k$.
5:         $\hat{w}_{F,k}^t \leftarrow TOA(w_{F,k}^t, s, l_k)$.                               ▷ Algorithm 2
6:         Send $\hat{w}_{F,k}^t$ and $w_{A,k}^t$ to $k$.
7:     **each** $k \in C_t$ **in parallel does**:
8:         $w_k^t \leftarrow \hat{w}_{F,k}^t \circ w_{A,k}^t$.
9:         For local epochs $1, ..., E$:
10:         $w_{A,k}^{t+1} = w_{A,k}^t - \eta \nabla f_k'(w_{A,k}^t)$.                        ▷ SGD
11:         Upload $w_{A,k}^{t+1}$ to the server.
12:     **for each layer** $W_l \in w^t$, **the server does**:
13:         $C_{t,l} \leftarrow \{k : k \in C_t \wedge W_l \in w_{A,k}^{t+1}\}$.        ▷ Obtain all clients that include $W_l$
14:         $n_l \leftarrow \sum_{k \in C_{t,l}} n_k$.
15:         $W_l \leftarrow \mathbb{E}(W_{k,l}) := \sum_{k \in C_{t,l}} \frac{n_k}{n_l} W_{k,l}$.       ▷ Layer-wise aggregation
16: **end for**
17: **return** $w^t$

---

## 3.2 FedOLF: Federated Learning with Ordered Layer Freezing

For a client $k$, the architecture of model $w$ can be decomposed into two components $w_{F,k}$ and $w_{A,k}$ such that $w = w_{F,k} \circ w_{A,k}$. $w_{F,k} = \{W_1, ..., W_{l_k}\}$ and $w_{A,k} = \{W_{l_k+1}, ..., W_N\}$ are respectively the set of frozen and active layers. $l_k \in \{0, 1, ..., N-1\}$ is the number of frozen layers in training whose value depends on $k$'s device capacity. For a powerful device that can train the entire model, we have $l_k = 0$ and $w_{F,k} = \varnothing$.

At global iteration $t$, client $k$ downloads the global model $w^t$ and decomposes $w^t$ into $w_{F,k}^t$ and $w_{A,k}^t$ based on $l_k$. Afterwards, client $k$ freezes $w_{F,k}^t$ and locally trains all parameters in $w_{A,k}^t$ by applying stochastic gradient descent (SGD) on dataset $D_k$ according to Equation (2):

$$w_{A,k}^{t+1} = w_{A,k}^t - \eta \nabla f_k'(w_{A,k}^t) \tag{2}$$

$\eta$ is the learning rate and $\nabla f_k'$ is a low-error-rate approximation of the gradient $\nabla f_k$ in the case of layer freezing. With layer freezing, the layers in $w_{F,k}^t$ will remain constant as training goes on, and will subsequently generate a straggling feature representation $\boldsymbol{x}_{l_k}' = \boldsymbol{x}_{l_k} + \boldsymbol{\sigma}_{l_k}$. $\boldsymbol{x}_{l_k}$ is the true representation generated by $w_{F,k}^t$ if it is non-freezing, and $\boldsymbol{\sigma}_{l_k}$ is an error term representing the divergence between $\boldsymbol{x}_{l_k}$ and $\boldsymbol{x}_{l_k}'$. Feeding $\boldsymbol{x}_{l_k}$ and $\boldsymbol{x}_{l_k}'$ forward will respectively result in $\nabla f_k$ and $\nabla f_k'$. Once local training is completed, client $k$ only sends the updated layers $w_{A,k}^{t+1}$ to the server for communication efficiency. After receiving the results from all participating clients, the server updates the global model using a layer-wise aggregation strategy same as in (Pfeiffer et al., 2023b). The details of FedOLF are outlined in Algorithm 1.

## 3.3 FedOLF with Tensor Operation Approximation

Furthermore, we propose an adapted Tensor Operation Approximation (TOA) framework (Adelman et al., 2021) dedicated to reducing the communication cost in FedOLF. Instead of the entire global

model $w$, a client $k$ downloads $\hat{w}_{F,k}^t$ and $w_{A,k}^t$ from the server, where $\hat{w}_{F,k}^t$ is a low-rank approximation of the frozen layers $w_{F,k}^t$ with fewer parameters. Unlike the initial TOA method which works on all layers, in this paper, the modified TOA works only on the frozen layers to ensure all active layers get fully trained. For illustration, let $H_q$ denote the number of tensors in a frozen layer $W_q$, where a tensor is a filter or neuron if $W_q$ is a convolution or fully-connected layer, respectively.

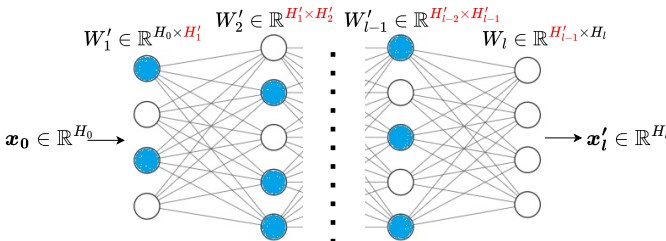

Figure 2: Within each frozen fully-connected layer $W_q$ ($1 \leq q < l$) containing $H_q$ neurons, a subset $W_q'$ (blue neurons) is derived by sampling $H_q' = \lfloor sH_q \rfloor$ neurons of the layer. Consequently, the approximation of $w_{F,k}^t$, represented as $\hat{w}_{F,k}^t$ is $\hat{w}_{F,k}^t = W_1' \circ ...... \circ W_{l-1}' \circ W_l$.

For example, Figure 2 shows how TOA is applied on a fully-connected neural network with $l$ frozen layers. For every layer $W_q$ ($1 \leq q < l$), except for the last frozen layer, the server samples $\lfloor sH_q \rfloor$ tensors from the layer and sends this subset of tensors to client $k$. $s$ ($0 < s \leq 1$) is a scaling factor that determines the trade-off degree between accuracy and communication efficiency, with $s = 1$ representing that no TOA is applied. Moreover, TOA is not performed on the last frozen layer as shown in Figure 2, so that the dimensions of the output representation $\boldsymbol{x}_l'$ and the following active layers remain unchanged. Based on the study of Adelman et al. (2021), we apply a weighted sampling strategy on TOA. With this strategy, TOA selects a tensor $\boldsymbol{Z}_j$ ($1 \leq j \leq H_q$) within a frozen layer $W_q$ with probabilities proportional to their *Frobenius* norms:

$$\mathbb{P}(\boldsymbol{Z}_j \in W_q') = \frac{\|\boldsymbol{Z}_j\|_F}{\sum_{j=1}^{H_q} \|\boldsymbol{Z}_j\|_F}. \tag{3}$$

In this case, the approximation error $\mathbb{E}[\|\boldsymbol{x}_l' - \boldsymbol{x}_{l,TOA}'\|^2]$ will be minimized, where $\boldsymbol{x}_{l,TOA}'$ and $\boldsymbol{x}_l'$ are respectively the output representations with and without TOA. The TOA technique significantly reduces the downstream communication cost in FedOLF by approximately $O(s^2)$. The procedure of TOA is shown in Algorithm 2.

---

**Algorithm 2** *TOA*

---

**Require:** set of frozen layers $w_F$, scaling factor $s$, number of frozen layers $l_k$:
  1: **For every layer** $W_q \in w_F, 1 \leq q \leq l_k - 1$:
  2:     $H_q \leftarrow len(W_q)$.
  3:     $W_q' \leftarrow$ sample(candidates=$\{Z_j\}_{j=1}^{H_q}$, weights=$\{\mathbb{P}(Z_j \in W_q')\}_{j=1}^{H_q}$, number=$\lfloor sH_q \rfloor$).
  4: **return** $w_F' := W_1' \circ ...... \circ W_{l_k-1}' \circ W_{l_k}$

---

### 3.4 DETERMINING THE NUMBER OF FROZEN LAYERS

Given a neural network $w$ with $N$ layers, the memory footprint $m(w)$ can be computed as:

$$m(w) = \sum_{q=1}^{N} m_{AM}(W_q) + m_G(W_q) + m_W(W_q) \approx \sum_{q=1}^{N} m_{AM}(W_q) \tag{4}$$

That is, the overall memory footprint $m$ is the accumulated memory footprint of three components, which are parameter weights ($m_W$), gradients ($m_G$) and activation maps ($m_{AM}$) across all layers. Moreover, compared with weights and gradients, the size of activation maps is much more massive

and consumes a dominant memory space. Therefore, the overall memory footprint can be approximated as the total size of activation maps across all layers (Pfeiffer et al., 2023a).

In FedOLF, for a frozen layer $W_q$, $m_{\text{AM}}(W_q)$ becomes zero, as no activation maps have to be stored for training (Pfeiffer et al., 2023a). Accordingly, a client $k$ can choose $l_k$ to be the smallest value, given $\sum_{q=l_k+1}^{N} m_{\text{AM}}(W_q)$ (the size of activation maps in the remaining active layers) not exceeding its memory limit.

### 3.5 LOW-LEVEL LAYER SHARING AMONG CLIENTS

According to the studies of (Zhang et al., 2024; Luo et al., 2021), low-level layers across various local models usually have higher degrees of Centered Kernal Alignment (CKA) similarity across different datasets (Kornblith et al., 2019), which means that these layers contain substantial redundant information and may generate similar feature representations. Motivated by this insight, in FedOLF, a resource-constrained device $k$ can "borrow" the highly-generalized low-level layers from other clients by downloading $w_{F,k}^t$ from the server. Layers in $w_{F,k}^t$ have been trained by more powerful clients in previous rounds, and can be directly employed by $k$ during the forward propagation phase of training without incurring significant errors.

### 3.6 VANISHING REPRESENTATION ERROR AND BOUNDED GRADIENT LOSS

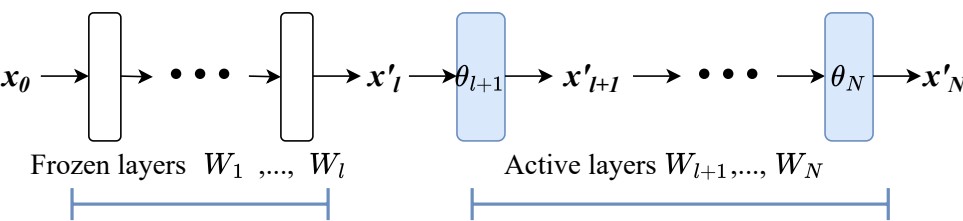

Figure 3: During training, the $l$ frozen layers will generate a feature representation $\boldsymbol{x}'_l$ that diverges from the true $\boldsymbol{x}_l$. Affected by $\boldsymbol{x}'_l$, the following active layers also generate inaccurate representations.

In addition to reducing memory usage, FedOLF preserves accuracy by mitigating representation errors induced by ordered layer freezing, with these errors diminishing as training advances through layers. For illustration, Figure 3 presents an exemplary model with $l$ frozen layers and $N - l$ active layers. As described in subsection 3.2, owing to the staleness of frozen layers, all feature representations after layer $W_l$ diverge from the true representations. However, the representation errors $\|\boldsymbol{\sigma}_l\|, \|\boldsymbol{\sigma}_{l+1}\|, ..., \|\boldsymbol{\sigma}_N\|$ tends to decrease as the depth grows, where $\|\cdot\|$ represents $l2$-norm.

To verify our hypothesis, we first make the following assumption:

**Assumption 1.** *The intrinsic function of each layer $W_l$ is $B_l$-Lipschitz continuous with $B_l > 0$:*

$$\forall \boldsymbol{x}_1, \boldsymbol{x}_2, \|W_l(\boldsymbol{x}_1, \boldsymbol{\theta}_l) - W_l(\boldsymbol{x}_2, \boldsymbol{\theta}_l)\| \leq B_l \|\boldsymbol{x}_1 - \boldsymbol{x}_2\|. \tag{5}$$

*Since $\boldsymbol{x}_{l+1} = W_l(\boldsymbol{x}_l, \boldsymbol{\theta}_l)$ and $\boldsymbol{x}'_{l+1} = W_l(\boldsymbol{x}'_l, \boldsymbol{\theta}_l)$, we can rewrite Equation (5) as:*

$$\|\boldsymbol{x}'_{l+1} - \boldsymbol{x}_{l+1}\| \leq B_l \|\boldsymbol{x}'_l - \boldsymbol{x}_l\|. \tag{6}$$

*By induction, we have:*

$$\|\boldsymbol{x}'_N - \boldsymbol{x}_N\| \leq \prod_{q=l}^{N-1} B_q \|\boldsymbol{x}'_l - \boldsymbol{x}_l\|, \text{ i.e. } \|\boldsymbol{\sigma}_N\| \leq \prod_{q=l}^{N-1} B_q \|\boldsymbol{\sigma}_l\|. \tag{7}$$

In the experiment, we find that the term $\prod_{q=l}^{N-1} B_q$ is always shrinking (see Appendix A for evidence). Consequently, the representation error $\|\boldsymbol{\sigma}_d\|$ ($l \leq d \leq N$) caused by layer freezing tends to be vanishing as $d$ increases, and the representation $\boldsymbol{x}'_d$ is gradually approaching the true representation $\boldsymbol{x}_d$, thereby narrowing the gap between the computed gradient $\nabla \boldsymbol{\theta}'_d$ and the true gradient $\nabla \boldsymbol{\theta}_d$. As a result, the accumulated training error $\sum_l^N \|\nabla \boldsymbol{\theta}'_l - \nabla \boldsymbol{\theta}_l\|$ will be bounded (see Appendix B.1).

## 4 CONVERGENCE ANALYSIS

In this section, we analyze the convergence results for FedOLF on non-convex smooth objective functions. We do not require the objective function to be convex in the case of deep-learning neural networks (Karimireddy et al., 2020). We make the following assumptions:

**Assumption 2** (smoothness). *The objective function $f_k$ is L-smooth:*

$$\forall w_1, w_2, \ \|\nabla f_k(w_1) - \nabla f_k(w_2)\| \leq L\|w_1 - w_2\|. \tag{8}$$

**Assumption 3** (Bounded variance). *The variance of local gradients to the global gradient is bounded:*

$$\forall k, w, \ \mathbb{E}(\|\nabla f_k(w) - \nabla f(w)\|^2) \leq \gamma^2. \tag{9}$$

Furthermore, from Assumption 1 and Assumption 2, we can infer that the divergence of local gradient $\|\nabla f_k' - \nabla f_k\|$ resulting from layer freezing is bounded, which is defined in Corollary 1.

**Corollary 1**. *For any client $k$, the divergence between the local gradient with and without layer freezing is bounded:*

$$\forall k, w, \ \|\nabla f_k'(w) - \nabla f_k(w)\|^2 \leq D^2. \tag{10}$$

Based on Assumptions 1-3 and Corollary 1, we derive the following theorems:

**Theorem 1**. *When the learning rate $\eta$ satisfies $\frac{1}{L} < \eta < \frac{3}{2L}$, we have:*

$$f(w^{t+1}) - f(w^t) \leq$$
$$\frac{\eta}{2}(2\eta L - 3)(\mathbb{E}[\|\nabla f(w^t)\|])^2 + \eta D(\eta L - 1)\mathbb{E}[\|\nabla f(w^t)\|] + \frac{\eta}{2}(2\eta L\gamma^2 - \gamma^2 + \eta LD^2 + 2\eta LD\gamma). \tag{11}$$

**Theorem 2**. *When the learning rate $\eta$ satisfies $\eta \leq \frac{1}{L}$, we have:*

$$f(w^{t+1}) - f(w^t) \leq \frac{\eta}{2} \times (-\mathbb{E}[\|\nabla f(w^t)\|^2] + D^2 + \gamma^2 + 2D\gamma). \tag{12}$$

According to Theorem 1 and Theorem 2, when the learning rate is less than $\frac{3}{2L}$, the objective function $f$ continues to decrease before $w^t$ reaching a $\epsilon$-*critical point* where $\|\nabla f(w^t)\| \leq \epsilon$. Specifically, when $\frac{1}{L} < \eta < \frac{3}{2L}$, we have $\epsilon = \epsilon_1 = \frac{D(\eta L-1)+\sqrt{\eta D^2 L+8\eta L\gamma^2+6\eta DL\gamma+D^2-3\gamma^2}}{3-2\eta L}$. When $\eta \leq \frac{1}{L}$, we have $\epsilon = \epsilon_2 = D + \gamma$. The proof can be found in Appendix B.

For FedOLF with TOA, the above theorems remain valid. The only difference is that the boundary $D$ in Corollary 1 is expected to become larger as TOA slightly increases the representation error. Subsequently, the critical points $\epsilon_1$ and $\epsilon_2$ also increase, resulting in an earlier halt in the decay of $f$.

## 5 EXPERIMENTS

### 5.1 EXPERIMENT SETUP

**Datasets and models.** We evaluate the performance of FedOLF on the Extended MNIST (EMNIST) (Cohen et al., 2017), CIFAR-10 (Krizhevsky et al., 2009), CIFAR-100 (Krizhevsky et al., 2009) and CINIC-10 (Darlow et al., 2018) datasets. For EMNIST, we adopt a convolutional neural network (CNN) consisting of two convolution layers and one fully-connected (FC) classifier (Horváth et al., 2021). For CIFAR-10, we employ AlexNet (Krizhevsky et al., 2012) (five convolution layers + two FC layers). For CIFAR-100 and CINIC-10, we utilize ResNet20 and ResNet44 (He et al., 2016).

**State-of-the-art for comparison.** We compare FedOLF with the following representative methods for resource-constrained FL: **1. Federated Dropout (Feddrop)** (Caldas et al., 2018) randomly prunes tensors in the global model and sends the remaining sub-model to clients for training. **2. FjORD** (Horváth et al., 2021) prunes the rightmost tensors of the global model. **3. HeteroFL** (Diao et al., 2021) prunes the rightmost filters in convolution layers similar to FjORD, but keep the FC layers unchanged. **4. DepthFL** (Kim et al., 2023) applies a top-first layer pruning method, and adds extra classifiers to clients with fewer layers to distill knowledge. **5. CoCoFL** (Pfeiffer et al., 2023b) let all clients store a full model locally and randomly freeze layers in training. **6. Successive**

| Dataset | | **EMNIST** | **CIFAR-10** | **CIFAR-100** | | **CINIC-10** | |
| --- | --- | --- | --- | --- | --- | --- | --- |
| Model | | CNN | AlexNet | ResNet20 | ResNet44 | ResNet20 | ResNet44 |
| Feddrop | | 32.11 | 14.33 | 17.02 | 6.2 | 9.87 | 10.31 |
| FjORD | | 7.55 | 46.3 | 12.7 | 14.68 | 16.55 | 20.08 |
| HeteroFL | | 17.4 | 54.79 | 12.32 | 12.96 | 10.69 | 10.03 |
| DepthFL | | 60.25 | 16.74 | 24.87 | 37.82 | 9.97 | 34.28 |
| CoCoFL | | 83.71 | 61.83 | 22.16 | 27.56 | 25.66 | 26.67 |
| SLT | | 60.72 | 30.47 | 25.04 | 43.73 | 24.11 | 33.63 |
| **FedOLF** | no TOA | **84.02** | **68.27** | **37.85** | **48.15** | **32.27** | **35.57** |
| | TOA(0.75) | - | 66.6 | 36.04 | 40.72 | 31.85 | 32.52 |
| | TOA(0.5) | - | 63.12 | 24.93 | 29.68 | 31.92 | 30.89 |
| FedAvg | | 84.42 | 69.22 | 46.01 | 49.46 | 36.32 | 37.59 |

Table 1: Comparison of the final test accuracy (in %) for $T = 500$ iterations in the non-iid case. Note that for EMNIST where the number of frozen layers is at most one, FedOLF+TOA is not evaluated as TOA only works with at least two frozen layers.

**Layer Training (SLT)** (Pfeiffer et al., 2023a) mandates all clients to sequentially train each layer from bottom to top, while freezing the parameters of the remaining layers. We also include the standard **FedAvg** benchmark (McMahan et al., 2017) for reference. All methods have run for three independent trials with their mean performance being recorded.

**Parameter settings and system implementation.** The experiment runs on a virtual network consisting of $K = 100$ clients operating on a desktop computer with one NVIDIA GeForce GTX 1650 GPU. The number of participants per round is $|C_t| = 10$ following the settings in (Horváth et al., 2021; Caldas et al., 2018). The maximum global iteration is set to $T = 500$ and the local training epoch is $E = 5$ for all clients (Horváth et al., 2021; Li et al., 2021a). The learning rate is set to $\eta = 0.0001$ for EMNIST, $\eta = 0.001$ for CIFAR-10, and $\eta = 0.01$ for CIFAR-100 and CINIC-10 (Luo et al., 2021). The batch size is set to 16 for EMNIST and 128 for the remaining datasets (Li et al., 2021a; Horváth et al., 2021). The experiment is implemented with PyTorch 2.0.0 and Flower 1.4.0(Beutel et al., 2022).

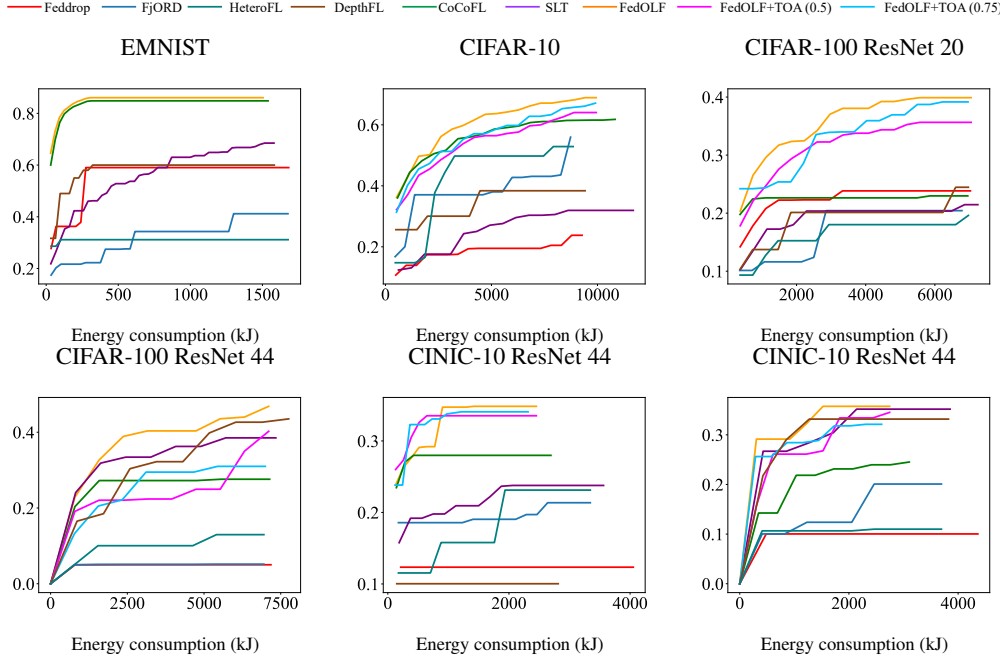

Figure 4: The curves of top-1 accuracy vs. energy consumption (kJ).

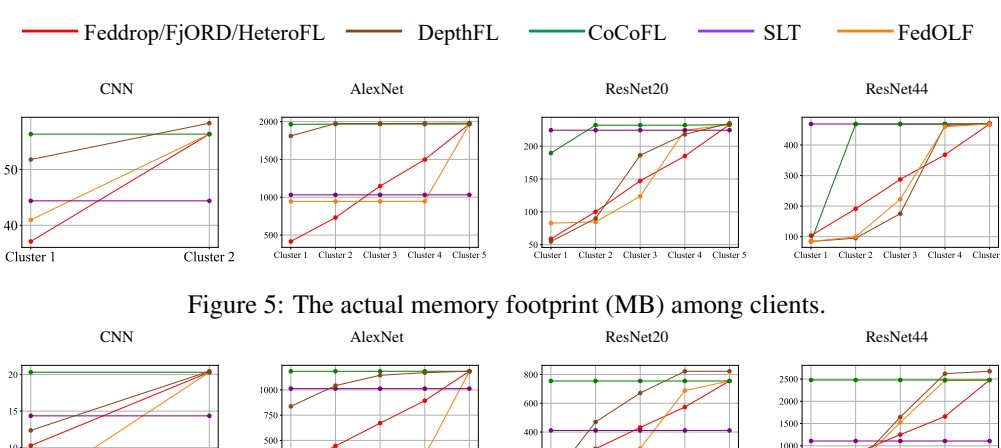

Figure 5: The actual memory footprint (MB) among clients.

Figure 6: The theoretical context-independent memory footprint (MB) among clients.

**Data and system heterogeneity.** We evaluate FedOLF in both iid and non-iid environments. For the iid case, data are allocated to clients uniformly. For the non-iid case, we follow (Luo et al., 2021) and allocate data to clients based on an extreme Dirichlet distribution with parameter 0.1. To emulate system heterogeneity, we divide all clients into $c$ uniform clusters that represent $c$ different degrees of device capability and resource constraints, as per (Horváth et al., 2021; Diao et al., 2021; Kim et al., 2023; Pfeiffer et al., 2023b). Specifically, following (Horváth et al., 2021), for CNN on EMNIST, $c$ is set to 2, wherein the numbers of pruned/frozen layers are 0 and 1 respectively for DepthFL, CoCoFL and FedOLF; for Feddrop and FjORD, the sub-model ratios (i.e. the percentage of left neurons per layer) are 0.5 and 1.0 for each cluster; for AlexNet on CIFAR-10 or ResNet20 on CIFAR-100/CINIC-10, $c = 5$ and the sub-model ratios are $\{0.2, 0.4, 0.6, 0.8, 1.0\}$ for Feddrop/FjORD; and the number of pruned/frozen layers or blocks are $\{4, 3, 2, 1, 0\}$ for DepthFL, CoCoFL, and FedOLF (see Appendix D for details). For SLT that conducts universal successive training on all clients, the scaling factor for the partial training procedure is set to 0.5 (Pfeiffer et al., 2023a).

## 5.2 EXPERIMENT RESULTS

**Accuracy.** Table 1 shows the accuracy comparison in the non-iid case (see Appendix C for the iid case). As shown in Table 1, FedOLF achieves the highest final accuracy among all methods on all datasets, which demonstrates the strength of FedOLF in preserving accuracy on resource-constrained devices. By looking through all methods, we find that dropout (Feddrop, FjORD) performs poorly with non-iid data, as training a sub-model cannot extract sufficient knowledge from the local dataset to construct an accurate global model (Pfeiffer et al., 2023b). Besides, sub-models with inconsistent architectures usually learn divergent parameter updates in training, and aggregating these updates altogether will inevitably compromise the global model's performance (Jiang et al., 2022). Although existing layer freezing approaches (CoCoFL, SLT) improve accuracy by maintaining the full model architecture on all clients, it still lags behind FedOLF in accuracy. Because in CoCoFL or SLT, the gradient loss caused by frozen layers does not decay as in FedOLF, and impedes performance more straightforwardly.

**Energy consumption and overall efficiency.** Combining accuracy and energy consumption, we can derive the energy efficiency of each method. As shown in Figure 4, FedOLF significantly improves energy efficiency by obtaining the highest accuracy with the same amount of energy expenditure. The specific computation and communication costs, including FLOPs, data transmission volume and energy, can be found in Appendix C.

**Memory footprint.**[1] We measure the real maximum memory usage of all methods using the `TORCH.CUDA.MAX_MEMORY_ALLOCATED` function (PyTorch, 2023). Moreover, considering that

---

[1]We merge the curves of Feddrop, FjORD, HeteroFL for brevity as their memory footprints are very close.

the real memory usage is usually context-dependent (physical device, programming language, etc), we also calculate their theoretical memory usage following Equation (4). As shown in Figures 5 and 6, FedOLF effectively reduces the memory footprint both theoretically and practically.

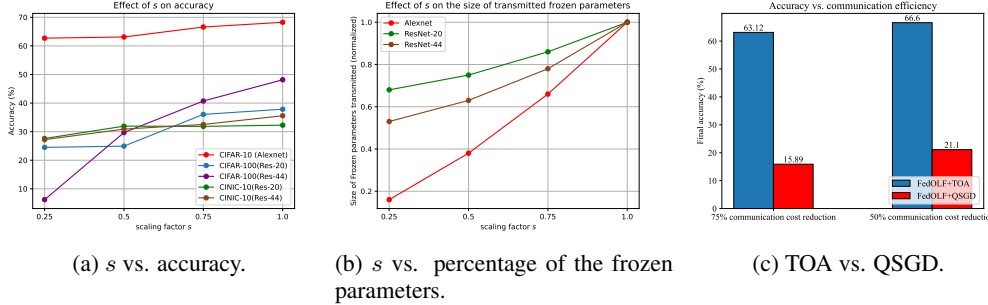

(a) $s$ vs. accuracy.

(b) $s$ vs. percentage of the frozen parameters.

(c) TOA vs. QSGD.

Figure 7: Effect of TOA and the scaling of factor $s$.

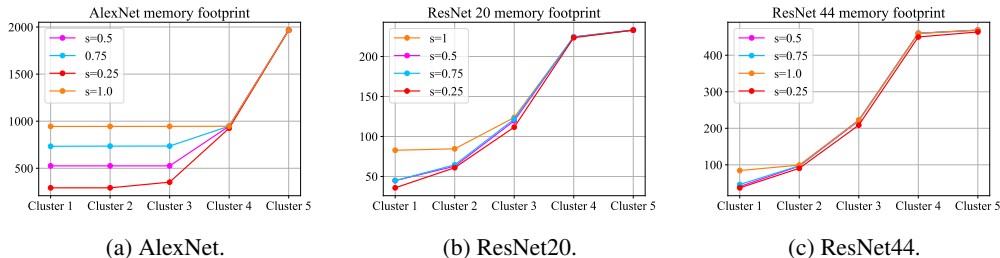

(a) AlexNet.

(b) ResNet20.

(c) ResNet44.

Figure 8: Effect of the scaling of factor $s$ on the practical memory footprint (MB).

**Hyperparameter tuning and ablation study.** We tune the scaling factor of TOA $s$ using a grid search within $\{0.25, 0.5, 0.75, 1.0\}$, where $s = 1$ is equivalent to FedOLF without TOA. Results in Table 1 and Figures 7a and 7b reveal that TOA effectively reduces the downstream communication cost without degrading much accuracy (except for CIFAR-100 with ResNet-44). For example, a scaling factor $s = 0.25$ can reduce the size of the transmitted frozen parameters by utmost 84% with a minor 5.56% accuracy loss compared with FedOLF sole (AlexNet). Besides, TOA further reduces the practical memory footprint as Figure 8 shows. Additionally, we compare TOA with the well-recognized quantized SGD (QSGD) method (Alistarh et al., 2017) for AlexNet on CIFAR-10. As shown in Figure 7c, TOA achieves much higher accuracy than QSGD given the same degree of communication efficiency. Specifically, TOA ($s = 0.5$) is compared with QSGD with 8 bits and TOA ($s = 0.75$) is compared with QSGD with 16 bits so that their reductions of communication cost are approximately equal.

## 6 CONCLUSION

This paper proposed Federated Learning with Ordered Layer Freezing (FedOLF), an efficient FL framework where edge devices only train the top-level layers of the model to accommodate resource constraints. The OLF strategy can minimize the backpropagation path length and the gradient error, which significantly reduces the memory requirement and improves accuracy. We also enhance FedOLF with the Tensor Operation Approximation (TOA) technique (Adelman et al., 2021), further alleviating energy consumption and memory footprint with less accuracy sacrifice. In the future, we plan to explore the similarities of local clients by using techniques like learning vector quantization (Qin & Suganthan, 2005), to make similar clients share the same layer freezing and TOA settings. We also plan to enhance the engagement of FedOLF in IoT applications such as mobile edge networks (Jin et al., 2024) and video surveillance (Zhang et al., 2022a).

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

## A    VANISHING REPRESENTATION ERROR AND BOUNDED TRAINING LOSS

We find that in FedOLF, the negative impact of low-level frozen layers tends to vanish. In formulation, let $\sigma_{l,l+1} \in \mathbb{R}^+$ denote the ratio between the representation error between two consecutive frozen layers $W_l$ and $W_{l+1}$, that is:

$$\sigma_{l,l+1} := \frac{\|\boldsymbol{\sigma}_{l+1}\|}{\|\boldsymbol{\sigma}_l\|} = \frac{\|\boldsymbol{x}'_{l+1} - \boldsymbol{x}_{l+1}\|}{\|\boldsymbol{x}'_l - \boldsymbol{x}_l\|} = \frac{\|W_{l+1}(\boldsymbol{x}'_l, \boldsymbol{\theta_{l+1}}) - W_{l+1}(\boldsymbol{x}'_l, \boldsymbol{\theta_{l+1}})\|}{\|\boldsymbol{x}'_l - \boldsymbol{x}_l\|}. \tag{13}$$

In addition, we rewrite Assumption 1 here for better illustration:

**Assumption 1.** *The intrinsic function of each layer $W_l$ is $B_l$-Lipschitz continuous with $B_l > 0$:*

$$\forall \boldsymbol{x}_1, \boldsymbol{x}_2, \|W_l(\boldsymbol{x}_1, \boldsymbol{\theta}_l) - W_l(\boldsymbol{x}_2, \boldsymbol{\theta}_l)\| \le B_l \|\boldsymbol{x}_1 - \boldsymbol{x}_2\|. \tag{14}$$

*By induction:*

$$\|\boldsymbol{\sigma}_N\| \le \prod_{q=l}^{N-1} B_q \|\boldsymbol{\sigma}_l\|. \tag{15}$$

Empirically, we find that the accumulative product $\|\boldsymbol{\sigma}_N\| \le \prod_{q=l}^{N-1} B_q \|\boldsymbol{\sigma}_l\|$ across layers is usually *shrinking*. For example, for AlexNet on CIFAR-10, we compute the error ratios among all convolution layers for a randomly selected client who only freezes the first layer, as shown in Figure 9:

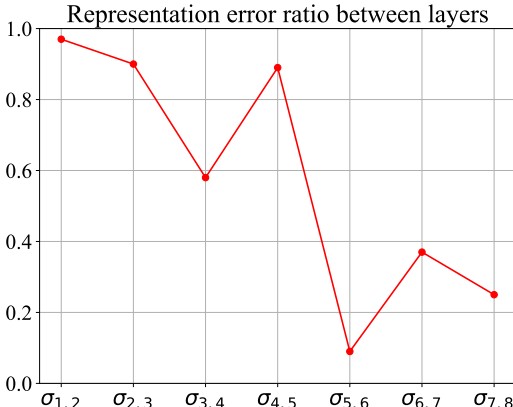

Figure 9: The ratios of the representation error between two consecutive layers in AlexNet **with only the first layer frozen**. Layers 1-5 are convolution layers, layers 6 and 7 are fully-connected layers and layer 8 is the classifier.

In this scenario, each boundary $B_l$ is highly likely to be smaller than one, which indicates that the representation error $\|\boldsymbol{\sigma}_{l+d}\| = \prod_{q=l}^{l+d-1} B_q \|\boldsymbol{\sigma}_l\|$ tends to vanish as $d$ increases. Consequently, the top level learns relatively accurate parameter updates by forwarding representations with lower error rates. Empowered by this property, FedOLF is able to achieve higher accuracy compared to other layer freezing methods.

For models like ResNet, the error ratios across layers are not consistently less than 1 as Figure 10 illustrates. We attribute this phenomenon to the unique architecture of ResNet, i.e. it adds connections between residual blocks so that the representation errors vanish less slowly. However, the term $\prod_{q=1}^{l-1} B_q$ still exhibits an overall vanishing trend as $l$ increases, because the remaining bounds $B_q$ for $q > 2$ are likely to be less than one. To further support the validity of this assumption, please refer to (Mirzasoleiman et al., 2020), where an equivalent assumption has also been made.

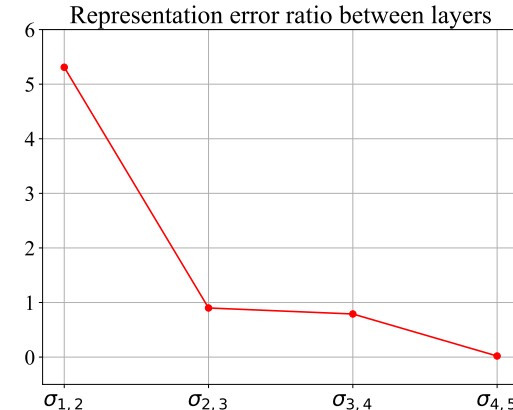

Figure 10: The ratios of the representation error between two consecutive residual blocks in ResNet20 **with only the first layer frozen**.

## B    THEORETICAL PROOF

This section presents the detailed proof of Corollary 1 and Theorems 1 and 2 in Section 4. First, we rewrite Assumptions 2 and 3 here:

**Assumption 2** (smoothness). *The local objective function $f_k$ is L-smooth:*

$$\forall w_1, w_2, \ \|\nabla f_k(w_1) - \nabla f_k(w_2)\| \leq L\|w_1 - w_2\|. \tag{16}$$

**Assumption 3** (Bounded variance). *The variance of local gradients to the global gradient is bounded:*

$$\forall k, w, \ \mathbb{E}(\|\nabla f_k(w) - \nabla f(w)\|^2) \leq \gamma^2. \tag{17}$$

### B.1    PROOF OF COROLLARY 1

Based on Assumptions 1 and 2, we can derive that the gradient divergence caused by layer freezing is bounded, as defined in Corollary 1:

**Corollary 1**. *For any client $k$, the divergence between the local gradient with and without layer freezing is bounded:*

$$\forall k, w, \ \|\nabla f'_k(w) - \nabla f_k(w)\|^2 \leq D^2. \tag{18}$$

*Proof.*

We can represent a model $w$ in the format of the set of all layers' parameters, i.e.:

$$w := (\boldsymbol{\theta}_1, \boldsymbol{\theta}_2, ..., \boldsymbol{\theta}_N). \tag{19}$$

Accordingly, for the gradient $\nabla f_k(w)$ we have:

$$\nabla f_k(w) = \nabla \boldsymbol{\theta}_1 + \nabla \boldsymbol{\theta}_2 + ...... + \nabla \boldsymbol{\theta}_N. \tag{20}$$

where $\nabla \boldsymbol{\theta}_l = \nabla f_k(\boldsymbol{x}_{l-1}, \boldsymbol{\theta}_l)$ for any $l$ ($1 \leq l \leq N$).

Furthermore, we define $\nabla \boldsymbol{\theta}'_l := \nabla f_k(\boldsymbol{x}'_{l-1}, \boldsymbol{\theta}_l)$. Then we can use $\|\nabla \boldsymbol{\theta}'_l - \nabla \boldsymbol{\theta}_l\|$ to represent the gradient error on the $l$-th layer caused by layer freezing.

Since $f_k$ is $L$-smooth, we have:

$$\|\nabla \boldsymbol{\theta}'_l - \nabla \boldsymbol{\theta}_l\| = \|\nabla f_k(\boldsymbol{x}'_{l-1}, \boldsymbol{\theta}_l) - \nabla f_k(\boldsymbol{x}_{l-1}, \boldsymbol{\theta}_l)\| \leq L\|\boldsymbol{x}'_{l-1} - \boldsymbol{x}_{l-1}\| = L\|\boldsymbol{\sigma}_{l-1}\|. \tag{21}$$

By induction, we have:

$$\|\nabla \boldsymbol{\theta}'_{l+d} - \nabla \boldsymbol{\theta}_{l+d}\| \leq L\|\boldsymbol{\sigma}_{l+d-1}\| \leq \prod_{q=l}^{l+d-1} B_q L\|\boldsymbol{\sigma}_l\|. \tag{22}$$

Based on Equations (20) and (22), for the gradient difference $\|\nabla f_k' - \nabla f_k\|$, we have:

$$
\begin{aligned}
&\|\nabla f_k'(w) - \nabla f_k(w)\| \\
&= \|\sum_{l=1}^{N} \nabla \boldsymbol{\theta}_l' - \sum_{l=1}^{N} \nabla \boldsymbol{\theta}_l\| \\
&= \|\sum_{l=1}^{N} (\nabla \boldsymbol{\theta}_l' - \nabla \boldsymbol{\theta}_l)\| \\
&\leq \sum_{l=1}^{N} \|\nabla \boldsymbol{\theta}_l' - \nabla \boldsymbol{\theta}_l\| \\
&\leq \sum_{l=2}^{N} \prod_{q=l}^{l-1} B_q L \|\boldsymbol{\sigma}_1\|.
\end{aligned}
\tag{23}
$$

Note that the first term $\|\nabla \boldsymbol{\theta}_1' - \nabla \boldsymbol{\theta}_1\|$ equals zero and gets eliminated in Equation (23), because $\nabla \boldsymbol{\theta}_1' = \nabla \boldsymbol{\theta}_1 = \mathbf{0}$ when the number of frozen layers is at least one.

Given that $\prod_{q=l}^{l-1} B_q$ is gradually vanishing as shown in Appendix A, the summation $\sum_{l=2}^{N} B^{l-1} L \|\boldsymbol{\sigma}_1\|$ must be finite and can be upper-bounded, alogn with $\|\nabla f_k'(w) - \nabla f_k(w)\|$. Therefore, Corollary 1 is naturally proven by setting $D$ as the boundary.

### B.2 PROOF OF THEOREM 1 AND THEOREM 2

Based on Assumptions 1-3 and Corollary 1, we derive Theorem 1 and Theorem 2:

**Theorem 1**. *When the learning rate $\eta$ satisfies $\frac{1}{L} < \eta < \frac{3}{2L}$, we have:*

$$
\begin{aligned}
&f(w^{t+1}) - f(w^t) \\
&\leq \frac{\eta}{2}(2\eta L - 3)(\mathbb{E}[\|\nabla f(w^t)\|])^2 + \eta D(\eta L - 1)\mathbb{E}[\|\nabla f(w^t)\|] + \frac{\eta}{2}(2\eta L \gamma^2 - \gamma^2 + \eta L D^2 + 2\eta L D \gamma).
\end{aligned}
\tag{24}
$$

**Theorem 2**. *When the learning rate $\eta$ satisfies $\eta \leq \frac{1}{L}$, we have:*

$$
f(w^{t+1}) - f(w^t) \leq \frac{\eta}{2} \times (-\mathbb{E}[\|\nabla f(w^t)\|^2] + D^2 + \gamma^2 + 2D\gamma).
\tag{25}
$$

*Proof.*

Since every $f_k$ is $L$-smooth based on Assumption 2, $f$ is also $L$-smooth, so that we have:

$$
f(w^{t+1}) - f(w^t) \leq \langle w^{t+1} - w^t, \nabla f(w^t) \rangle + \frac{L}{2}\|w^{t+1} - w^t\|^2.
\tag{26}
$$

In the setting of layer freezing, we have $w^{t+1} = w^t - \eta \nabla f'(w^t)$ and $\nabla f'(w^t) = \mathbb{E}[\nabla f_k'(w^t)]$. Therefore:

$$
\begin{aligned}
&f(w^{t+1}) - f(w^t) \\
&\leq -\eta \langle \mathbb{E}[\nabla f_k'(w^t)], \nabla f(w^t) \rangle + \frac{L}{2}\| - \eta \mathbb{E}[\nabla f_k'(w^t)]\|^2 \\
&= -\eta \mathbb{E}[\langle \nabla f_k'(w^t), \nabla f(w^t) \rangle] + \frac{L\eta^2}{2}\|\mathbb{E}[\nabla f_k'(w^t)]\|^2 \\
&\leq -\eta \mathbb{E}[\langle \nabla f_k'(w^t), \nabla f(w^t) \rangle] + \frac{L\eta^2}{2}\mathbb{E}(\|\nabla f_k'(w^t)\|^2).
\end{aligned}
\tag{27}
$$

Since $\|\nabla f'_k(w^t) - \nabla f(w^t)\|^2 = \|\nabla f'_k(w^t)\|^2 - 2\langle \nabla f'_k(w^t), \nabla f(w^t)\rangle + \|\nabla f(w^t)\|^2$, Equation (27) can be written as:

$$f(w^{t+1}) - f(w^t)$$

$$\leq \frac{\eta}{2}\mathbb{E}(\|\nabla f'_k(w^t) - \nabla f(w^t)\|^2 - \|\nabla f'_k(w^t)\|^2 - \|\nabla f(w^t)\|^2) + \frac{L\eta^2}{2}\mathbb{E}(\|\nabla f'_k(w^t)\|^2)$$

$$= \frac{\eta}{2}\mathbb{E}(\|\nabla f'_k(w^t) - \nabla f(w^t)\|^2) + \frac{\eta}{2}(\eta L - 1)\mathbb{E}[\|\nabla f'_k(w^t)\|^2] - \frac{\eta}{2}\mathbb{E}[\|\nabla f(w^t)\|^2] \qquad (28)$$

$$= \frac{\eta}{2}\mathbb{E}\|(\nabla f'_k(w^t) - \nabla f_k(w^t) + \nabla f_k(w^t) - \nabla f(w^t)\|^2)$$

$$+ \frac{\eta}{2}(\eta L - 1)\mathbb{E}[\|\nabla f'_k(w^t)\|^2] - \frac{\eta}{2}\mathbb{E}[\|\nabla f(w^t)\|^2].$$

According to Cauchy-Schwarz inequality, $\|\nabla f'_k(w^t) - \nabla f_k(w^t) + \nabla f_k(w^t) - \nabla f(w^t)\|^2 \leq (\|\nabla f'_k(w^t) - \nabla f_k(w^t)\| + \|\nabla f_k(w^t) - \nabla f(w^t)\|)^2$. Therefore, from Equation (28) we get:

$$f(w^{t+1}) - f(w^t)$$

$$\leq \frac{\eta}{2}\mathbb{E}[(\|\nabla f'_k(w^t) - \nabla f_k(w^t)\| + \|\nabla f_k(w^t) - \nabla f(w^t)\|)^2]$$

$$+ \frac{\eta}{2}(\eta L - 1)\mathbb{E}[\|\nabla f'_k(w^t)\|^2] - \frac{\eta}{2}\mathbb{E}[\|\nabla f(w^t)\|^2]$$

$$= \frac{\eta}{2}\mathbb{E}(\|\nabla f'_k(w^t) - \nabla f_k(w^t)\|^2 + \|\nabla f_k(w^t) - \nabla f(w^t)\|^2) \qquad (29)$$

$$+ 2\mathbb{E}(\|\nabla f'_k(w^t) - \nabla f_k(w^t)\| \times \|\nabla f_k(w^t) - \nabla f(w^t)\|)$$

$$+ \frac{\eta}{2}(\eta L - 1)\mathbb{E}[\|\nabla f'_k(w^t)\|^2] - \frac{\eta}{2}\mathbb{E}[\|\nabla f(w^t)\|^2]$$

$$\leq \frac{\eta}{2}(D^2 + \gamma^2 + 2D\gamma) + \frac{\eta}{2}(\eta L - 1)\mathbb{E}[\|\nabla f'_k(w^t)\|^2] - \frac{\eta}{2}\mathbb{E}[\|\nabla f(w^t)\|^2].$$

The last inequality in Equation (29) results from Corollary 1 and Assumption 3.

When the learning rate $\eta > \frac{1}{L}$, we have $\eta L - 1 > 0$. In this case, we can upper bound $\frac{\eta}{2}(\eta L - 1)\mathbb{E}[\|\nabla f'_k(w^t)\|^2]$ by upper bounding $\mathbb{E}[\|\nabla f'_k(w^t)\|^2]$.

First, we bound $\|\nabla f'_k(w^t)\|$. Based on Corollary 1 and triangle inequality, we have:

$$\|\nabla f'_k(w^t)\| - \|\nabla f_k(w^t)\| \leq \|\nabla f'_k(w^t) - \nabla f_k(w^t)\| \leq D. \qquad (30)$$

That is:

$$\|\nabla f'_k(w^t)\|^2 \leq (\|\nabla f_k(w^t)\| + D)^2 = \|\nabla f_k(w^t)\|^2 + D^2 + 2D\|\nabla f_k(w^t)\|. \qquad (31)$$

By taking the expectation on Equation (31), we get:

$$\mathbb{E}[\|\nabla f'_k(w^t)\|^2] \leq \mathbb{E}[\|\nabla f_k(w^t)\|^2] + D^2 + 2D\,\mathbb{E}[\|\nabla f_k(w^t)\|]. \qquad (32)$$

Because of the triangle inequality, we have:

$$\mathbb{E}[\|\nabla f_k(w^t)\|] = \mathbb{E}[\|\nabla f_k(w^t) - \nabla f(w^t) + \nabla f(w^t)\|]$$

$$\leq \mathbb{E}[\|\nabla f_k(w^t) - \nabla f(w^t)\|] + \mathbb{E}[\|\nabla f(w^t)\|] \qquad (33)$$

$$\leq \mathbb{E}[\|\nabla f(w^t)\|] + \gamma.$$

The last inequality in Equation (33) holds because $\mathbb{E}[\|\nabla f_k(w^t) - \nabla f(w^t)\|] \leq \gamma$ as $(\mathbb{E}[\|\nabla f_k(w^t) - \nabla f(w^t)\|])^2 \leq \mathbb{E}[\|\nabla f_k(w^t) - \nabla f(w^t)\|^2] \leq \gamma^2$ by Assumption 3. Moreover, by expanding Assumption 3, we have:

$$\mathbb{E}[\|\nabla f_k(w^t)\|^2]$$

$$= \mathbb{E}[\|\nabla f_k(w^t) - \nabla f(w^t) + \nabla f(w^t)\|^2]$$

$$= \mathbb{E}[\|\nabla f(w^t)\|^2] + \mathbb{E}[\|\nabla f_k(w^t) - \nabla f(w^t)\|^2] + 2\mathbb{E}(\langle \nabla f_k(w^t) - \nabla f(w^t), \nabla f(w^t)\rangle)$$

$$\leq \mathbb{E}[\|\nabla f(w^t)\|^2] + \gamma^2 + 2\mathbb{E}(\langle \nabla f_k(w^t) - \nabla f(w^t), \nabla f(w^t)\rangle) \qquad (34)$$

$$\leq \mathbb{E}[\|\nabla f(w^t)\|^2] + \gamma^2 + \mathbb{E}(\|\nabla f_k(w^t) - \nabla f(w^t)\|^2 + \|\nabla f(w^t)\|^2)$$

$$= 2\mathbb{E}[\|\nabla f(w^t)\|^2] + 2\gamma^2.$$

By combining Equations (32), (33), (34) altogether, we get:

$$
\begin{aligned}
&\mathbb{E}[\|\nabla f'_k(w^t)\|^2] \\
&\leq \mathbb{E}[\|\nabla f_k(w^t)\|^2] + D^2 + 2D\,\mathbb{E}[\|\nabla f_k(w^t)\|] \\
&\leq 2\mathbb{E}[\|\nabla f(w^t)\|^2] + 2\gamma^2 + D^2 + 2D\,\mathbb{E}[\|\nabla f_k(w^t)\|] \\
&\leq 2\mathbb{E}[\|\nabla f(w^t)\|^2] + 2\gamma^2 + D^2 + 2D(\mathbb{E}[\|\nabla f(w^t)\|] + \gamma) \\
&= 2\mathbb{E}[\|\nabla f(w^t)\|^2] + 2D\,\mathbb{E}[\|\nabla f(w^t)\|] + 2\gamma^2 + D^2 + 2D\gamma.
\end{aligned}
\tag{35}
$$

Accordingly, we can rewrite Equation (29) as:

$$
\begin{aligned}
&f(w^{t+1}) - f(w^t) \\
&\leq -\frac{\eta}{2}\mathbb{E}[\|\nabla f(w^t)\|^2] + \frac{\eta}{2}(D^2 + \gamma^2 + 2D\gamma) + \frac{\eta}{2}(\eta L - 1)\mathbb{E}[\|\nabla f'_k(w^t)\|^2] \\
&\leq -\frac{\eta}{2}\mathbb{E}[\|\nabla f(w^t)\|^2] + \frac{\eta}{2}(D^2 + \gamma^2 + 2D\gamma) \\
&\quad + \frac{\eta}{2}(\eta L - 1) \times (2\mathbb{E}[\|\nabla f(w^t)\|^2] + 2D\,\mathbb{E}[\|\nabla f(w^t)\|] + 2\gamma^2 + D^2 + 2D\gamma) \\
&= \frac{\eta}{2}(2\eta L - 3)\mathbb{E}[\|\nabla f(w^t)\|^2] + \eta D(\eta L - 1)\mathbb{E}[\|\nabla f(w^t)\|] + \frac{\eta}{2}(2\eta L\gamma^2 - \gamma^2 + \eta L D^2 + 2\eta L D\gamma).
\end{aligned}
\tag{36}
$$

When $2\eta L - 3 < 0$, i.e. $\eta < \frac{3}{2L}$, we have $(2\eta L - 3)\mathbb{E}[\|\nabla f(w^t)\|^2] \leq (2\eta L - 3)(\mathbb{E}[\|\nabla f(w^t)\|])^2$. In this case, Equation (29) can be written as:

$$
\begin{aligned}
&f(w^{t+1}) - f(w^t) \\
&\leq \frac{\eta}{2}(2\eta L - 3)(\mathbb{E}[\|\nabla f(w^t)\|])^2 + \eta D(\eta L - 1)\mathbb{E}[\|\nabla f(w^t)\|] + \frac{\eta}{2}(2\eta L\gamma^2 - \gamma^2 + \eta L D^2 + 2\eta L D\gamma).
\end{aligned}
\tag{37}
$$

Which successfully proves Theorem 1. Furthermore, if we take $\mathbb{E}[\|\nabla f(w^t)\|]$ as a **variable**, $f(w^{t+1}) - f(w^t)$ is deemed to be upper bounded by a **polynomial function** of $\mathbb{E}[\|\nabla f(w^t)\|]$. In this case, we can naturally find $\epsilon_1 = \frac{-b-\sqrt{b^2-4ac}}{2a}$ by letting the polynomial function equal to zero, with $a = 2\eta L - 3$, $b = 2D(\eta L - 1)$ and $c = 2\eta L\gamma^2 - \gamma^2 + \eta L D^2 + 2\eta L D\gamma$.

After calculation, we can get $\epsilon_1 = $

$$
\frac{D(\eta L - 1) + \sqrt{\eta D^2 L + 8\eta L\gamma^2 + 6\eta D L\gamma + D^2 - 3\gamma^2}}{3 - 2\eta L}
\tag{38}
$$

Similarly, when the learning rate $\eta \leq \frac{1}{L}$, we have $\eta L - 1 \leq 0$. In this case, $\frac{\eta}{2}(\eta L - 1)\mathbb{E}[\|\nabla f'_k(w^t)\|^2]$ is naturally upper bounded by zero, so that Equation (29) can be written as:

$$
f(w^{t+1}) - f(w^t) \leq \frac{\eta}{2}(D^2 + \gamma^2 + 2D\gamma) - \frac{\eta}{2}\mathbb{E}[\|\nabla f(w^t)\|^2].
\tag{39}
$$

Which successfully proves Theorem 2. By letting $\frac{\eta}{2}(D^2 + \gamma^2 + 2D\gamma) - \frac{\eta}{2}\mathbb{E}[\|\nabla f(w^t)\|^2]$ equal to zero we naturally get $\epsilon_2 = D + \gamma$.

## C  SUPPLEMENTRAY EXPERIMENT RESULTS

### C.1  COMPUTATION AND COMMUNICATION OVERHEADS

The single-sided comparison of computation cost (in FLOPs) and communication cost (in size of data transmission) are shown in Figures 11 and 12.

### C.2  ENERGY CONSUMPTION

The overall energy consumptions, including the computation energy for training, and the communication energy for parameter transmission, are shown in Figure 13. The energy consumption is measured using a plug-in power monitor [2].

---

[2] https://www.amazon.com.au/Electricity-Monitor-PIOGHAX-Overload-Protection/dp/B09SFSB66M

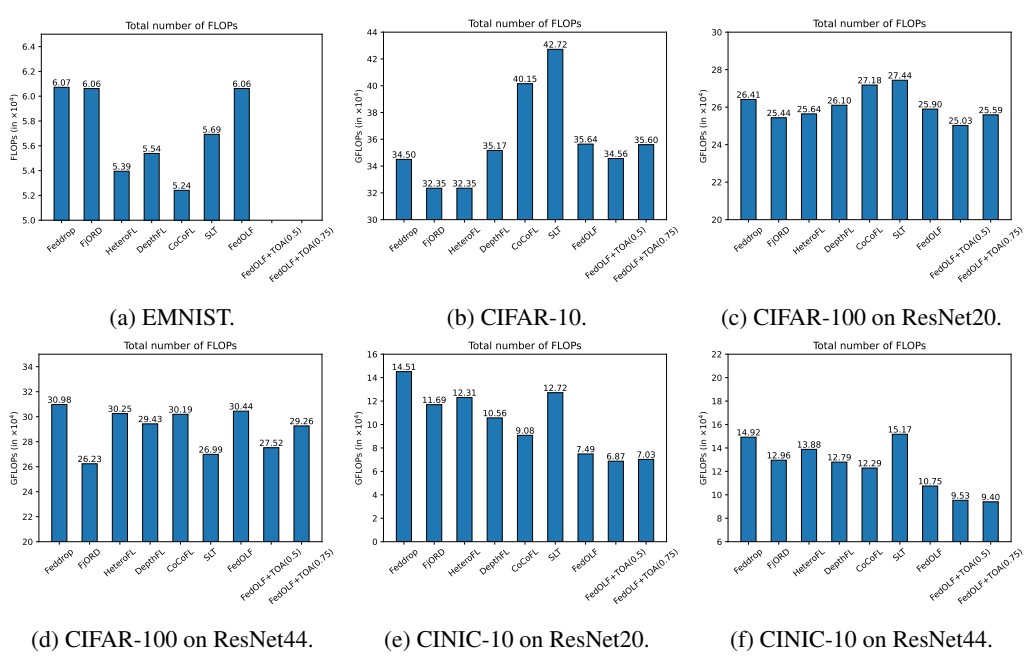

Figure 11: Comparison of the overall computation cost, which is measured in total Floating Point Operations (FLOPs) of all clients. This is the average result for the three trials.

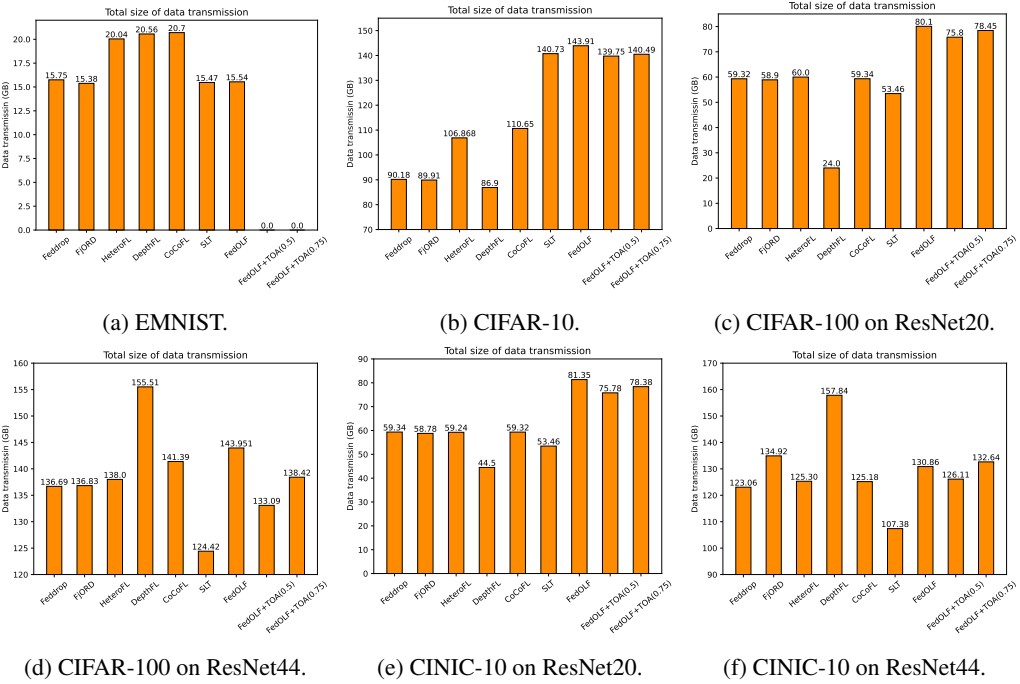

Figure 12: Comparison of the overall communication cost, which is measured in the total size of parameters transmitted across the network. This is the average result for the three trials.

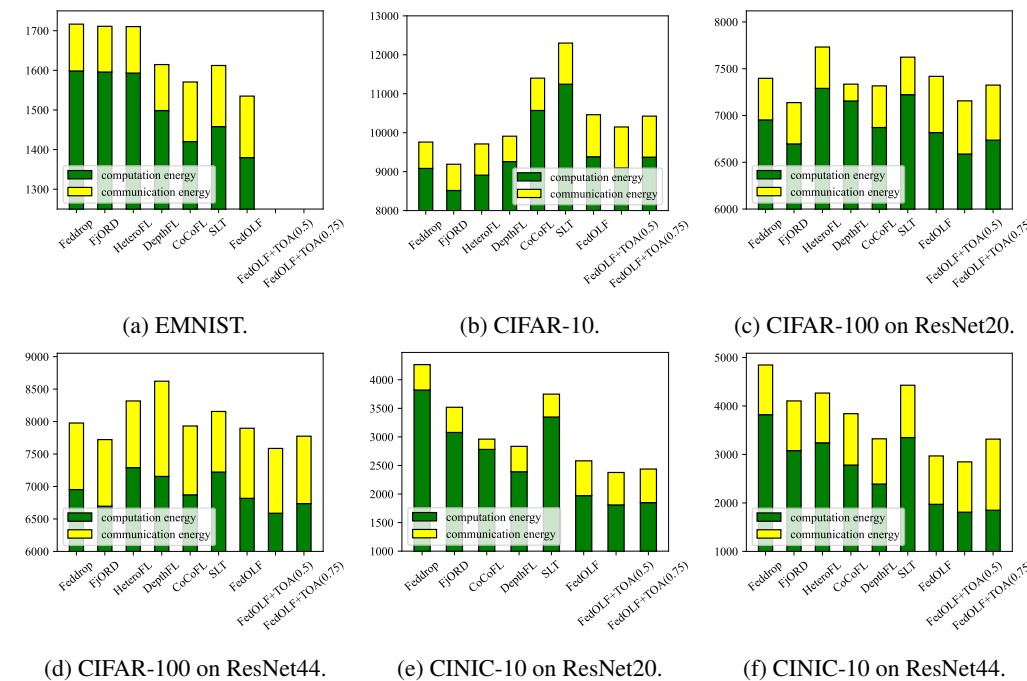

(a) EMNIST.  (b) CIFAR-10.  (c) CIFAR-100 on ResNet20.

(d) CIFAR-100 on ResNet44.  (e) CINIC-10 on ResNet20.  (f) CINIC-10 on ResNet44.

Figure 13: An overview of the overall energy consumption (kJ) of all clients, including the computation energy for local training (green) and the communication energy for global communication (yellow).

| Dataset | | **EMNIST** | **CIFAR-10** | **CIFAR-100** | | **CINIC-10** | |
|---|---|---|---|---|---|---|---|
| Model | | CNN | AlexNet | ResNet20 | ResNet44 | ResNet20 | ResNet44 |
| Feddrop | | 16.42 | 14.33 | 6.05 | 6.15 | 9.71 | 10.81 |
| FjORD | | 12.68 | 27.8 | 11.14 | 9.09 | 22.22 | 11.26 |
| HeteroFL | | 12.88 | 58.03 | 7.02 | 14.32 | 13.46 | 12.0 |
| DepthFL | | 83.0 | 10.52 | 5.05 | 39.88 | 10.31 | 33.44 |
| CoCoFL | | 81.98 | 53.92 | 26.95 | 31.1 | 31.81 | 31.68 |
| SLT | | 81.04 | 49.73 | 45.80 | 39.60 | 21.63 | 36.20 |
| **FedOLF** | no TOA | **84.98** | **66.98** | **48.49** | **44.12** | **40.66** | **37.33** |
| | TOA(0.75) | - | 63.7 | 40.49 | 42.16 | 33.96 | 31.51 |
| | TOA(0.5) | - | 62.05 | 36.19 | 38.29 | 33.42 | 28.42 |
| FedAvg | | 85.04 | 68.41 | 51.11 | 52.13 | 40.80 | 39.88 |

Table 2: Comparison of the final test accuracy (in %) for $T = 500$ iterations in the iid case.

## C.3 ACCURACY IN THE IID CASE

The accuracy comparison in the iid case is listed in Table 2. As shown in Table 2, FedOLF still outperforms the baselines in the iid case, and maintains a competitive accuracy against the FedAvg benchmark.

## C.4 ACCURACY VS. ROUND

The curves of accuracy with respect to training rounds are shown in Figures 14 and 15.

# D DEMONSTRATION OF LAYER FREEZING IN FEDOLF

Figures 16a, 16b, 16c illustrate how FedOLF freezes layers among the heterogeneous clients. Each bar represents a cluster of clients, dividing the model into two segments. On one side of the bar, denoted by "F," clients freeze the corresponding layers, while on the other side, denoted by "T," clients actively train the model. Specifically, clients within Cluster 2 for EMNIST and Cluster 5 for CIFAR-10/CIFAR-100/CINIC-10 are assumed to possess the capability to train the entire model. It is important to note that convolution layers are typically followed by activation functions (e.g., ReLU), batch normalization or max-pooling layers, though these are not depicted in the figures for brevity.

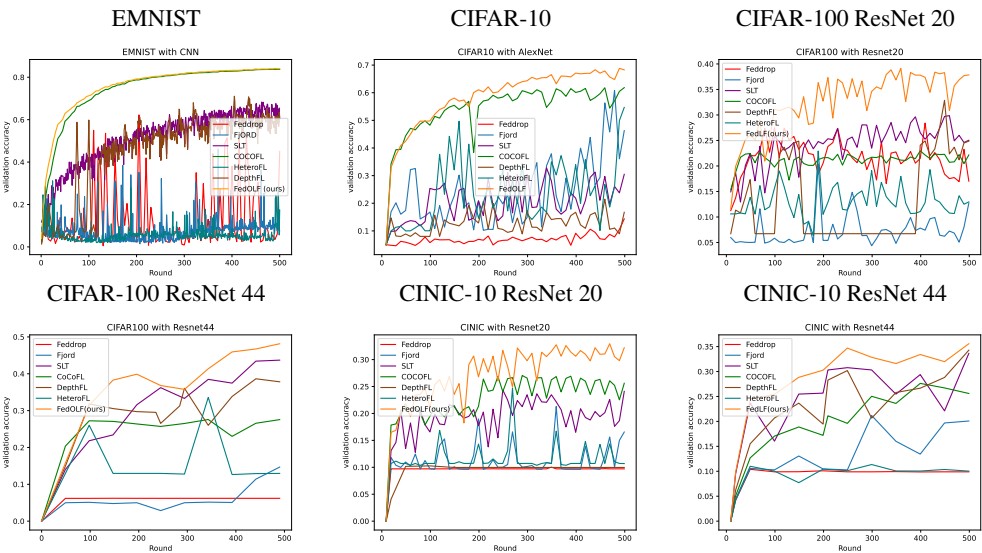

Figure 14: Accuracy vs. round in the non-iid case.

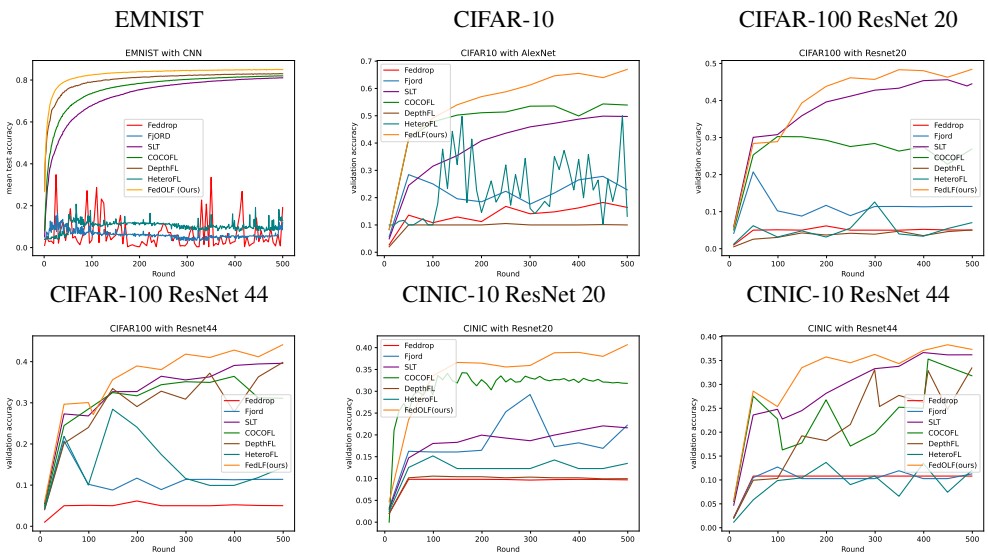

Figure 15: Accuracy vs. round in the iid case.

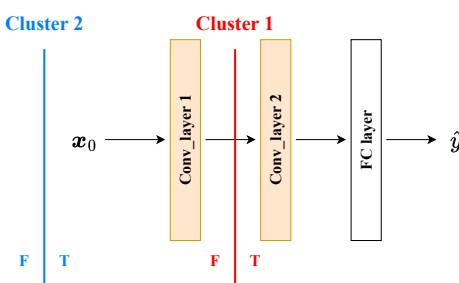

(a) Layer freezing pattern of the 2-layer CNN on EMNIST.

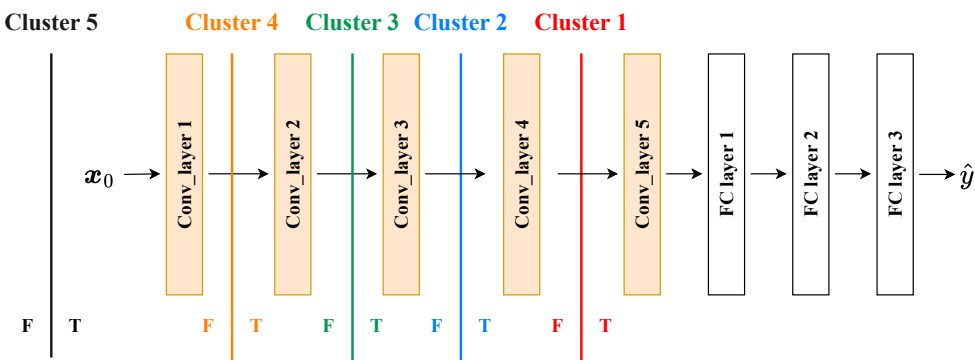

(b) Layer freezing pattern of AlexNet on CIFAR-10.

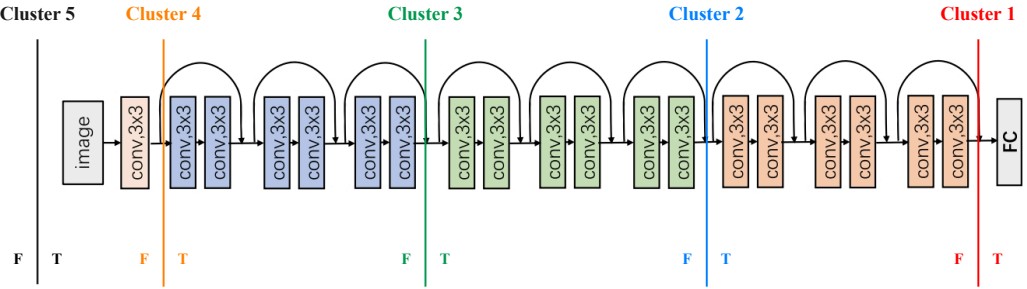

(c) Layer freezing pattern of ResNet20 on CIFAR-100 and CINIC-10. The figure of the network architecture is downloaded from Chen et al. (2022b). For ResNet44 with more layers per block, the same division rule also applies.

Figure 16: A specific demonstration of how FedOLF freezes layers among the heterogeneous clients.

# E  DISCUSSION

## E.1  LIMITATIONS

One of the major limitations of this paper is the lack of theoretical support of TOA's application on FedOLF. Even though TOA seems to work well based on the obvious reduction of energy consumption/memory footprint as shown in the experiment results, the specific relationship between performance degradation and the scaling factor $s$ remains unexplored. In this case, the only way to determine the optimal value of $s$ is through an experiment (i.e. hyperparameter tuning), which is usually computationally expensive. To enhance the usefulness of FedOLF w. TOA in practical applications, a close-formed representation of the effect of $s$ on accuracy is required, so that we can determine the optimal value of $s$ given the particular energy budget and accuracy requirement.

The other limitation of FedOLF is the additional communication overhead as shown in Figure 12. Even with TOA, the communication overhead of FedOLF is still higher than other methods in most cases. In the environment of our experiment, the connection between clients and the server is relatively stable, so that the extra communication overhead of FedOLF does not generate too much energy consumption. However, in a real-world system with underprivileged network conditions, such as a mobile-edge network Jin et al. (2024), the negative impact of the increased communication overhead becomes severe, resulting in much higher communication costs. To promote the application of FedOLF in bandwidth-constrained systems, addressing the concern of increased communication cost becomes a vital matter.

## E.2  BROADER IMPACT

FedOLF has a positive social impact on boosting fair FL training among heterogeneous clients. FedOLF proposes that powerful clients take more responsibility in training (i.e. train more layers), and share the low-level layers with weak clients for forwardpropagation in their local training tasks. This significantly improves FL's accuracy, efficiency and robustness in resource-constrained settings.

Furthermore, FedOLF alleviates privacy concerns compared with traditional FL frameworks such as Fedavg McMahan et al. (2017). As resource-constrained clients only communicate the active layers with the server. Compared with the full model, transmitting partial active layers reduces the risks of several types of attacks such as byzantine attack and privacy inference Hao et al. (2021); Cao et al. (2020); Zhang et al. (2022b).

As for the negative impact, the increased communication overhead might restrict the usefulness of FedOLF in mobile networking systems with insufficient bandwidth support. Except for TOA, possible solutions to address this problem include: **1. Periodical downward communication:** Clients download the frozen layers periodically rather than every round. **2. Clustering:** Clients download from a proximal header rather than the remote server.

