# OpenReview forum: "Energy and Memory-Efficient Federated Learning with Ordered Layer Freezing and Tensor Operation Approximation"
_ICLR.cc/2025/Conference — Submitted to ICLR 2025_

### Official Review · Reviewer_XVEh · 2024-11-04

**Soundness:** 2
**Presentation:** 3
**Contribution:** 2
**Rating:** 5
**Confidence:** 3

**Summary:**

The paper proposes the FedOLF framework that aims to improve the efficiency of FL for resource constrained devices. The key observation behind the work is that by allowing the update of the layers of a network in an ordered manner, energy and memory costs can be reduced without significant sacrifices in the performance of the system. The authors move further and enhance the efficiency of their system by employing Tensor Operation Approximation demonstrating better results in their setting compared to traditional quantization approaches.

One of the main claimed contributions of the work, which is also the basis of the method, is the vanishing representation error. That is by imposing an ordering of the layers that are frozen in any update, the authors demonstrate a bound on the representation error of the active layers as a function of the last frozen layers.

**Strengths:**

The idea (and proof) of bounding  the approximation of the true gradient through layer freezing under certain condition is important and can be used for the design of topologies that operate under such regime.

The experimental results show significant performance gains in the specific benchmark tests.

**Weaknesses:**

The idea of tackling the energy and memory consumption in FL by frozen a number of layers is not new and many works have already utilised this approach. The novel claimed contribution is the strategy of the ordering of the updating. Given that it is well known that more high level layers are more important for specific tasks compared to the early layers that have the job to extract more generic features, such approach is not so surprising.

The memory footprint can also be tackled by re-evaluating part of the results in an on-demand manner. As such, someone can trade off time for training vs memory footprint. Given that for many devices the computational cost (in terms of energy) is less than the communication cost, it is unclear how such approach would compare on the overall energy cost vs quality of training with the proposed method that chose to freeze some of the layers for addressing the memory requirements.

A key observation/assumption that underpins the theoretical working of the method is that the upper bound B is likely smaller that one. However this is only based on few networks and datasets and I found difficult to justify the above statement

In the experimental section, it was not clear to me what is the assumption of the quality of the training of the original network. For example, I would expect ordered layer freezing to perform well if the original model was not very far (and especially the initial layers) from the true one, where if you give to the system a network with random parameters (extreme case) then the initial layers would not be very informative and any imposed freezing on them would not lead to meaningful training for the rest of the layers. The authors should comment on that.

**Questions:**

it is not clear to me how the communication energy consumption is captured in this experiment. How do the authors provide their relative weights?

Regarding Table 1. There is no evidence that the final achieved accuracy would be better in the proposed system compared to the rest. Is the target number of iterations that give this result? How the performance of the above algorithms change with the iteration number T?

Someone would expect the benefits of the method to become more profound as the number of layers in the network increases. However, Table 1 shows that this is not the case and the gap between the proposed method and STL closes (even for T=500). Can the authors comment on that?

---

> ### Author Response · Authors · 2024-11-20
> **Response to weakness 1**
>
> Dear reviewer,
>
> Thank you for your insightful feedback. Below, we address each point raised in your comment:
>
> __For weakness 1__, we appreciate the reviewer’s observation regarding the prevalence of layer freezing in machine learning. While
> it is true that layer freezing has been explored extensively in centralized machine learning, our contribution specifically addresses its application in the FL paradigm, which presents unique challenges and dynamics not encountered in centralized settings. To the best of our knowledge, our work is the first to propose Ordered Layer Freezing (OLF) in FL, emphasizing the systematic ordering of layer updates based on task-specific importance—a key differentiator from existing works.

---

> > ### Comment · Reviewer_XVEh · 2024-11-25
> >
> > Thanks for your response. You stated that the contribution is on addressing the unique challenges of the FL paradigm. However, it is not clear from the paper what are those challenges compared to the centralised approach that your approach actually addresses. Can you please be more specific?
> >
> > You also mention a systematic ordering of the layer updates based on the task. Can you please point to where in your experiments you adopt (and how) the strategy based on the task? It is expected that different problems would require different setting of the hyper-parameters but it is not clear to me how your proposed approach adapts to the task at hand.

---

> ### Author Response · Authors · 2024-11-20
> **Response to weakness 2**
>
> __For weakness 2__, we appreciate the reviewer’s suggestion regarding trading off time for training versus memory footprint through
> on-demand re-evaluation of results. To address this trade-off in the context of OLF, we analyze the interplay between communication efficiency, memory footprint, and training quality. As shown in Figure 7b, decreasing the TOA scaling factor $s$ significantly reduces the size of the transmitted frozen layers, thereby lowering memory and communication costs. However, as TOA becomes more aggressive (i.e., $s$ is smaller), the model’s performance deteriorates proportionally. This decline in performance demands more training iterations, ultimately leading to increased energy consumption, as demonstrated in Figure 4, where the energy cost per accuracy increment rises
> with aggressive TOA settings. Furthermore, Figure 4 highlights that FedOLF consistently achieves the highest communication and computation efficiency among all methods without requiring additional compromises in training quality. Given that our approach already addresses memory constraints effectively while maintaining high training performance, adopting on-demand re-evaluation would introduce unnecessary overhead and inefficiency.

---

> > ### Comment · Reviewer_XVEh · 2024-11-25
> >
> > You have misunderstood my point. Assume an edge device with infinite processing power but limited memory. You can restructure the computation of the training and evaluate the info for each layer's weight update as needed without having to store all the data from the previous layers. As such, you would have minimised the memory footprint without any compromise in the training quality or latency. The TOA introduces error in your signal and it is expected that the training quality would decrease, but this is not my point.

---

> ### Author Response · Authors · 2024-11-20
> **Response to weakness 3**
>
> __For weakness 3__, we appreciate the reviewer’s observation. To address this, we note that our analysis accommodates scenarios where $B$ is not universally smaller than one, such as in networks like ResNet where $B$ is not smaller than one (see Appendix A). In this scenario, even though $B$ is not universally smaller than one across all layers, the representation errors still exhibit a decaying trend toward high-level layers.
>
> To further support this argument, we refer to Eq (16) and Appendix B.2 in paper [1] where an equivalent assumption is made.
> This paper first shows that the overall gradient error is bounded by a linear function of the last layer's gradient error, i.e. $\|\nabla f - \nabla f'\| \leq c_{1} \|\nabla f^{(N)} - \nabla f'^{(N)} \| + c_{2}$. ($c_{1}, c_{2}$ are constants and $N$ is the total layer number). Then the paper assumes $\|\nabla f'^{(N)} - \nabla f^{(N)} \|$ is bounded to therefore bound $\|\nabla f - \nabla f'\|$. This assumption is equivalent to the vanishing representation error justification in this paper. To prove this, first note $\|\nabla f^{(N)} - \nabla f'^{(N)}\| \leq L \|\sigma_{N-1}\| = L \prod_{q=1}^{N-1}B_{q}\|\sigma_{1}\|$ due to the $L$-smoothness of $f$. Then, if $\|\nabla f^{(N)} - \nabla f'^{(N)}\|$ is bounded, $\prod_{q=1}^{N-1}B_{q}$ must be shrinking, meaning that the representation error must be vanishing. This is because, by contradiction, if the errors are non-vanishing, $\prod_{q=1}^{N-1}B_{q}$ will explode as $N$ increases and $\|\nabla f^{(N)} - \nabla f'^{(N)}\|$ cannot be bounded.
>
> 1. Mirzasoleiman, Baharan, Jeff Bilmes, and Jure Leskovec. ”Coresets for data-efficient training of machine
> learning models.” In ICML, 2020.

---

> > ### Comment · Reviewer_XVEh · 2024-11-25
> >
> > Thank you for this explanation. The provided analysis is welcome but I would like to point out that there are still a number of assumptions in place in order for this to hold. My main point is that you make a statement about how likely is the upper bound to be smaller than 1, which is not backed up by experiments. I would be happy to see this claim backed up by experiments, or otherwise to change the claim and condition the results on the cases where the upper bound is less than 1.

---

> ### Author Response · Authors · 2024-11-20
> **Response to weakness 4**
>
> __For weakness 4__, to clarify, the effectiveness of FedOLF does not depend on the quality of the initial model parameters, either theoretically or empirically. Firstly, our convergence analysis in Section 4 does not rely on any assumptions related to model initialization. In FedOLF, given any arbitrary original point, the global model will eventually converge to a ϵ-critical point with bounded error. Secondly, in the experiment, FedOLF is always evaluated with random model initialization and still produces high accuracy.
> Therefore, the original model’s quality does not have any effect on FedOLF either theoretically or empirically.

---

> > ### Comment · Reviewer_XVEh · 2024-11-25
> >
> > Thanks for the clarification. However, the point is not only whether the model will be able to converge, but also on the rate of convergence. Can you please provide some data where you have repeated the experiments multiple times with random initial values for the parameters of the models and show the distribution of the convergence and the time that it takes?

---

> > > ### Author Response · Authors · 2024-12-03
> > > **Looking forward to the reviewer's feedback**
> > >
> > > Dear reviewer XVEh,
> > >
> > > We appreciate your effort in reviewing the paper and providing us with constructive feedback. We have written comprehensive responses with respect to your review.
> > >
> > > As the discussion period will end soon, please give us a comment on whether we have successfully addressed all your concerns. If so, we kindly hope you consider improving the rating.
> > >
> > > Regards

---

> ### Author Response · Authors · 2024-11-20
> **Response to all questions**
>
> __For Q1__, as described in Appendix C.2, the communication energy consumption is measured using a plug-in power monitor. In each communication round, we use the monitor to measure the energy consumption of the computer to acquire this round’s communication energy consumption, and the total communication cost is the summation of all rounds’ communication energy cost.
>
> __For Q2__, we thank the reviewer for raising this important point. We have updated our submission and attached the accuracy vs. round graphs to the appendix (Figs 14 and 15 on page 21). The graphs show that FedOLF (orange curve) has higher accuracy and better convergence in most cases.
>
> __For Q3__, we attribute this result to the unique property of SLT. For ResNet, SLT progressively trains each block in a bottom-first order. When the model increases from ResNet20 to ResNet44, the number of layers in each block also increases. In this case, the accuracy of SLT significantly increases as it trains more layers in the same rounds. Even though the performance gap between SLT and
> FedOLF gets narrowed as the model grows, FedOLF still achieves higher accuracy and energy efficiency as shown in Table 1 and Figure 4.
>
> Regards

---

> ### Author Response · Authors · 2024-11-27
> **For weakness 1**
>
> Dear reviewer,
>
> Thanks for your response.
>
> __For w1__, we apologize for the confusion. The unique challenge in FL is about __how to preserve accuracy under non-iid local data and resource-constrained edge devices__.
>
> In centralized ML, all data are stored in a central server. The server usually has a strong capacity and therefore a higher training cost. In this setting, layer freezing is applied to reduce the server's training cost. Since the powerful server usually has full-model training capacity, it can conduct a few full-training steps first to access the heuristic of all parameters, like weight or gradient. Then, the server can selectively freeze the unimportant parameters in each layer to reduce the computation cost with minimal accuracy loss. In centralized ML, accuracy and efficiency can be easily achieved simultaneously.
>
> In FL, however, devices with important data may not be able to train a full model owing to memory constraints, and therefore cannot participate in FL. When the global model cannot learn important knowledge from these devices, performance degradation will occur. To address this issue, we need to find a way to enable these important but memory-constrained devices to join FL.
>
> We provide an insightful analysis to show that ordered layer freezing (OLF) is the most effective method (compared with SOTA) to address this challenge with higher accuracy and smaller memory requirements, without using any heuristic (full-model training).
>
> For accuracy, prior works [1-3] show that high-level layers are more sensitive and customized to different local datasets. These layers need to be frequently trained to ensure that the global model learns sufficient knowledge from diverse local data. On the other hand, low-level layers are more generalized across different data distributions, i.e. they do not need to be trained by every client and can be shared among clients (we discussed this point in Section 3.5). Moreover, OLF largely preserves accuracy due to the vanishing representation error property (Section 3.6). Because of these advantages, FedOLF achieves higher accuracy than SOTA methods (see Table 1).
>
> For memory, we’ve shown that ordered-layer freezing significantly reduces memory footprint by minimizing the backward propagation path’s length (Fig. 1). Compared with existing layer freezing methods CoCoFL and SLT, FedOLF has much lower memory usage (Figs. 5-6) and hence is more compatible with memory-constrained edge devices.
>
> The expression “a systematic ordering of the layer updates based on the task” means that the hyperparameter $l_{k}$, i.e. the number of frozen layers , can be determined adaptively based on the concrete task. As shown in Section 3.4, in real scenarios, we can calculate the memory footprint based on the task’s specific content (e.g. what dataset / model / algorithm we use), and choose the proper $l_{k}$ to ensure that the memory usage does not exceed the limit of the physical device. Meanwhile, in the baselines, the hyperparameters are determined arbitrarily in the simulation, which is unformulated and difficult to use in real-world applications.
>
> [1] Weishan Zhang, Tao Zhou, Qinghua Lu, Yong Yuan, Amr Tolba, and Wael Said. Fedsl: A communication efficient federated learning with split layer aggregation. IEEE Internet of Things Journal, 2024.
>
> [2] Mi Luo, Fei Chen, Dapeng Hu, Yifan Zhang, Jian Liang, and Jiashi Feng. No fear of heterogeneity: Classifier calibration for federated learning with non-IID data. In NeurIPS, 2021.
>
> [3] Mirzasoleiman, Baharan, Jeff Bilmes, and Jure Leskovec. ”Coresets for data-efficient training of machine learning models.” In ICML, 2020.

---

> ### Author Response · Authors · 2024-11-27
> **For weaknesses 2-4**
>
> __For w2__, we believe that training time vs. memory tradeoff will not work well. First, we’ve shown that training lower layers isn’t quite memory-efficient (lines 55-86). For example, if you only train the first bottom layer while freezing the rest, the activation values in the frozen layers are needed to compute the gradient of the first layer, and still have to be stored in memory as discussed in [4]. Given that activation takes a great proportion of the overall memory usage, the memory usage of low-level layer training can be very close to full-model training. Therefore, for a device with strong computation resources but very limited memory space, training the first few layers will NOT take a long time for training, instead, it will raise the “out-of-memory” error and terminate immediately.
>
> On the contrary, in FedOLF, only the top active layers need to store the activation as lower frozen layers do not involve any gradient computation (see Fig. 1). Therefore, compared with other layer freezing schemes, FedOLF always minimizes the total activation size stored in memory given the same amount of frozen layers.
>
> Second, increasing the training time simultaneously increases the computation cost, which is not quite efficient. Also, as mentioned in Section 3.5, lower layers generalize well across different local datasets and can be shared between clients. Therefore, it is unnecessary for memory-constrained clients to train lower-layers, as they can directly use the layers that are pre-trained by other clients.
>
> __For w3__, we appreciate your constructive feedback. We have updated our paper and removed the “B<1” claim. As a replacement, we re-write Assumption 1 as “each layer $W_{l}$ is $B_{l}$-lipschitz continuous”. That is, we change $B$ to $B_{l}$, meaning that the bound is no longer universal and depends on the layers. We find that even though $B_{l}$ can be large occasionally for some layer in some particular model structures (e.g. ResNet), the accumulative product $\prod\_{l} B\_{l}$ usually presents a overall decaying trend (This claim is supported by the results in Appendix A). This means that the overall training error, which is proportional to $\sum^{N}\_{l} \prod^{l}\_{q}B_{q}$ is always bounded, and our convergence analysis still holds (see Appendix B).
>
> __For w4__, we have attached the graphs reflecting accuracy per iteration in Appendix D (Figs 14-15, page 21). In all scenarios, we use an random initial model with accuracy close to zero (i.e. the initial model is very far from the optimal model). The graphs show that FedOLF (orange curve) has higher accuracy and better convergence in most cases, regardless of the quality of the initial model.
>
> [4] Kilian Pfeiffer, Ramin Khalili, and Joerg Henkel. Aggregating capacity in FL through successive layer training for computationally-constrained devices. In NeurIPS, 2023a.
>
> Regards.

---

> ### Author Response · Authors · 2024-12-02
> **Discussion deadline approaching**
>
> Dear reviewer XVEh,
>
> It's been a couple of days since we replied to your latest response. As the discussion deadline is close, we would like you to provide a feedback as soon as possible.
>
> Regards

---

### Official Review · Reviewer_hgpf · 2024-11-04

**Soundness:** 2
**Presentation:** 3
**Contribution:** 2
**Rating:** 3
**Confidence:** 4

**Summary:**

The proposed approach, FedOLF, combines the layer freezing method and Tensor Operation Approximation (TOA) to reduce the memory and computation requirements of the training process in federated learning settings for the memory-constrained IoT devices/systems. Experimental results show better overall accuracy with low memory, and energy usage compared to prior works, for different datasets like EMNIST, CIFAR-10, CIFAR-100, and CINIC-10.

**Strengths:**

1.	Applying TOA to frozen layers preserves model accuracy while reducing the communication cost.

2.	The experimental results of FedOLF on different iid and non-iid datasets and architectures demonstrate its adaptability and advantage in both energy and accuracy over baseline methods.

**Weaknesses:**

1.	The paper offers limited novelty, as FedOLF's approach combines previously explored techniques, partial neural network freezing and TOA. The combination of existing techniques, although seems useful, requires more thorough evaluation to distinguish the scenarios and system configurations in which it outperforms the state-of-the-art techniques and scenarios and systems configurations in which it offers sub-optimal results. Specifically, how well the technique performs compared to state-of-the-art for system configurations in which most of the clients are resource constrained (in terms of memory and compute both)? Also, the paper should demonstrate the performance of the proposed technique for non-uniform clusters of devices. Additionally, the proposed technique should be evaluated for cases with a skewed distribution of classes. All these analyses will provide a detailed performance comparison with the state-of-the-art techniques and a better understanding of the significance of the novel contributions presented in the paper.

2.	FedOLF assumes that "low-level layers across various local models usually have higher degrees of Centered Kernel Alignment (CKA) similarity," allowing frozen layers to generalize across clients without retraining. While Table 1 shows strong accuracy compared to the benchmark models in non-IID settings, supporting FedOLF's effectiveness, it does not directly validate this assumption of universal feature similarity. In real-world scenarios with highly diverse data (e.g., medical vs. satellite images), this may not hold true, and the frozen layers will become less useful for some clients. To validate this assumption, the experimentation should include more types of non-IID characteristics, such as label distribution skew, feature distribution skew, and quantity skew across clients. Additionally, a theoretical analysis of the conditions under which CKA similarity holds or fails, as well as experiments for validating this similarity across different layers and datasets, would be useful. Moreover, it could be interesting to perform a more in-depth analysis of the performance of the individual layers when considering the heterogeneity of the data; it may offer more detailed insights into the practical limits of the proposed approach.

3.	The number of frozen layers is determined just based on the memory footprint of different options, and the computational requirements are not considered. The authors should highlight how this is scalable to systems composed of devices with diverse set of computational capabilities. Moreover, is considering just the memory footprint for real systems sufficient? If not, what modifications are required in the proposed technique, and how incorporating computational requirements into the layer freezing decision can improve the performance of the proposed technique for practical systems?

4.	By freezing specific low-level layers, FedOLF implicitly assumes that the representation error from these frozen layers will diminish as training progresses. While the paper claims that these errors vanish empirically, a detailed analysis of error propagation through frozen layers seems missing. To address this, a layer-by-layer analysis of error propagation throughout training, using visualizations like heat maps to track error dynamics, is required. Additionally, experiments that vary the number and position of frozen layers could provide crucial insights into how these changes impact error propagation and overall model performance.

**Questions:**

The overall contributions seem marginal and further evaluations are necessary to uncover the true potential and limitations of the proposed technique. Besides addressing the above-raised comments in the weakness section, following questions also need particular attention.

1.	What general guidelines or thresholds can be provided for setting the scaling factor s in TOA, beyond using a grid search, to balance accuracy and communication costs effectively? or maybe is there a way to determine the optimal value of s without using grid search?
2.	The TOA technique in FedOLF uses a fixed scaling factor across all clients, which might not fit well for devices with different processing power (different hardware). Therefore, it is important to explore a more flexible, client-specific scaling factor to improve FedOLF's efficiency on diverse IoT devices?
3.	To emulate system heterogeneity, why only the case of uniform clusters is considered?
4.	In the context of Section 3.5, how many clients with full network training capacity are necessary for this technique to offer reasonable (and fast) convergence? Also, can this technique be used when most of the devices are significantly memory constrained. If yes, what are the limits? Also, how does the proposed technique compare with the state-of-the-art techniques under such scenarios.
5.	How well the technique performs in cases with a skewed distribution of classes (and general categories) across devices? Are there any data-level assumptions that have to be valid for the technique to offer reasonable results?
6.	What is the impact of the proposed technique in cases with significant computational power imbalance between devices? What is the impact of such cases on the overall training time?

---

> ### Author Response · Authors · 2024-11-20
> **Response to weakness 1**
>
> Dear reviewer,
>
> Thank you for your insightful feedback. Below, we address each point raised in your comment.
>
> __For weakness 1__, as stated in the experiment setup (lines 400-408), the experiments are already conducted in a scenario where most (80\%) are resource-constrained and unable to train an entire model. In practice, the definition of "resource-constrained" can be complicated and multi-aspect (e.g. computation, communication and memory). In the experiment simulation, the resource constraints can be reflected by the number of frozen layers for each client as discussed in [4], and we follow the same setting here as justified in lines 460-461.
>
> For uniform clustering, this paper follows the same experiment setting of FjORD which uniformly divides clients into five clusters [1]. Besides, most existing works employ a uniform client clustering policy in the experiment settings. For example, HeteroFL [2] divides clients into five uniform clusters with sub-model ratios 0.0625, 0.125, 0.25, 0.5 and 1.0. DepthFL [3] has four uniform clusters with 1-4 residual blocks respectively. CoCoFL [4] has three uniform client clusters that train 1/3, 2/3, and all of the layers respectively. Therefore, uniform client clustering is a quite common and sufficient selection to simulate a real-world network with heterogeneous clients in FL (see our justification in lines 454-456). Furthermore, to the best of our knowledge, none of the existing works utilize a non-uniform clustering strategy. If you have specific references that demonstrate the significance of non-uniform clustering for federated learning, we would be eager to explore this further.
>
> For skewness, the evaluation has already been conducted with a skewed class distribution among clients. As introduced in the experiment setup, this paper follows [5] and allocates data to clients unevenly. Specifically, the total number of samples per client, and the number of categories per client both follow a Dirichlet Distribution with parameter $\alpha$. The smaller $\alpha$ is, the more skewed class distributions we have. In this paper, we set $\alpha=0.1$, which is already considered an extreme case for class imbalance according to [5] (see lines 452-454). Given this setup, we believe that our evaluation already captures the challenges posed by class imbalance. If you could provide more insight into why you think further evaluation with more skewed distributions is necessary and what kind of skewness setting can represent typical real-world scenarios, we would be happy to consider expanding the experiments.

---

> > ### Comment · Reviewer_hgpf · 2024-11-24
> > **Reviewer's Feedback on Authors' Response to Weakness 1 and Question 4**
> >
> > In response to Question 4, the authors have stated that "The trade-off is that if fewer clients have full-model training capability, the accuracy will be somewhat lower due to increased gradient errors D from frozen layers (see Section 4). However, the system will still converge to a solution, albeit with lower accuracy than if more clients were able to perform full training."
> > This statement clearly states the limitations of the proposed technique. In the context of the above statement, the remaining parts of Question 4 should also be answered. Remaining parts of question 4: "Also, can this technique be used when most of the devices are significantly memory constrained. If yes, what are the limits? Also, how does the proposed technique compare with the state-of-the-art techniques under such scenarios."
> >
> > The non-uniform clustering is just to identify the limitations of the proposed technique in the context where only a very limited number of devices (e.g., just one or none) have full model training capability. If state-of-the-art techniques outperform the proposed under such scenarios, what scenarios are ideal for the proposed technique?

---

> > ### Comment · Reviewer_hgpf · 2024-11-25
> >
> > The novelty of the paper, as per the authors' response to one of the reviewers' lies in ordered layer freezing concept. However, the authors have not tested the concept under harsh memory constraints. Moreover, computational and communication constraints can also play a significant role in the performance / effectiveness of the system and such constraints have been completely ignored by the authors. In response to the questions, the authors have provided just a few references to support their claim of memory being the most important constraint, when a lot of references can be stated that highlight considering all three together.
> >
> > Moreover, layer freezing and quantization have been proposed in COCOFL, reference [4] provided by the authors in their responses.

---

> ### Author Response · Authors · 2024-11-20
> **Response to weakness 2**
>
> We would like to clarify some points regarding the assumption about Centered Kernel Alignment (CKA) and its applicability in the FedOLF approach.
>
> First, we would like to clarify that FedOLF __does not__ "assume" that low-level layers across local models exhibit higher degrees of CKA similarity. Specifically, the conclusion that low-level layers tend to have higher CKA similarity is based on the theoretical results presented in prior works, notably in [5-6, 9].
>
>
> As stated in lines 279-282 of our paper, we rely on the findings of these works, which empirically and theoretically demonstrate that low-level layers across different models tend to exhibit higher CKA similarity than higher-level layers. Therefore, this is not an assumption, but rather a proven result derived from prior research.
>
> For a deeper theoretical understanding of why CKA is an appropriate metric for comparing the similarity of layer outputs, we encourage you to refer to Section 3 in [9], which provides a detailed explanation. It highlights why CKA is a robust metric due to its ability to capture important properties like non-invertible linear transformations and isotropic scaling, which are crucial for comparing feature representations in neural networks, we’ve also cited this work in lines 282-283.
>
> Regarding your concern about the performance of individual layers under diverse data distributions, we refer you to several works that provide empirical evidence of how low-level layers exhibit more consistent CKA similarity across various datasets and conditions. For example, Figs. 1-3 in [5], Figs 2-3 in [6], and Fig. 7 in [9]. These results show that lower layers universally obtain higher CKA similarities than upper layers across different datasets, non-iid data distributions, or model architectures. Therefore, we believe that the idea of low-level layer sharing will continuously work well even on highly diverse data, we have included this justification in lines 280-283.

---

> > ### Comment · Reviewer_hgpf · 2024-11-25
> >
> > The reliance on prior works is valid to some extent, but the concern about testing FedOLF with diverse non-IID data distributions remains poorly addressed. Specifically, the claim that "low-level layers universally obtain higher CKA similarity" is not directly validated with additional datasets or scenarios in this paper.

---

> ### Author Response · Authors · 2024-11-20
> **Response to weakness 3**
>
> We agree that both memory and computational constraints are crucial when considering the scalability of a federated learning approach. However, as discussed in [8], memory constraints are typically more critical than computational and communicational constraints in real-world federated learning scenarios. This is because memory limitations can directly prevent a device from participating in the federated learning process, whereas computational and communicational constraints mainly affect efficiency, slowing down the training process or increasing costs. In contrast, memory constraints are more restrictive, as
> a device unable to store sufficient model parameters cannot participate in training, leading to exclusion from the FL process altogether. This is especially problematic in non-IID environments, where excluding important clients can result in significant performance degradation, we introduce this point in lines 37-40.
>
> In addition, even though we do not particularly target addressing computational and communications constraints, our method still has higher computation and communication efficiency than existing methods due to its superior performance. As shown in Fig.4, FedOLF naturally improves the computation and communication efficiency by obtaining higher accuracy with the same amount of overall (computation+communication) energy consumption. Based on these points, we believe that considering the memory footprint is a sufficient design for FedOLF.

---

> > ### Comment · Reviewer_hgpf · 2024-11-25
> >
> > What happens if a device has limited computational capability and cannot complete its training task in time? The device is dropped for that round or it makes the whole system wait? Which policy was used during experimentation. Depending on the policy, the quality of output, and the skewness of data, the computational requirements can be highly relevant and cannot be ignored. Thus, all computational, memory, and communication constraints have to be considered for the system design. See [4].

---

> > ### Comment · Reviewer_hgpf · 2024-11-25
> >
> > The computational and communications factors are not properly addressed, memory constraints alone don't capture the complexity of real-world FL environments where computational power can vary significantly across devices.

---

> > > ### Author Response · Authors · 2024-12-02
> > > **Discussion deadline approaching**
> > >
> > > Dear reviewer hgpf,
> > >
> > > It's been a couple of days since we replied to your latest response. As the discussion deadline is close, we would like you to provide feedback as soon as possible.
> > >
> > > Regards

---

> ### Author Response · Authors · 2024-11-20
> **Response to weakness 4**
>
> Thank you for your insightful comment. We appreciate your suggestion to include a detailed layer-by-layer analysis of error propagation and to visually track error dynamics during training. Below, we address your concerns and provide additional clarification on the vanishing error assumption for frozen layers.
>
> For in-depth analysis of vanishing representation layers, please refer to Appendices A and B, in formula, the norm of error $\sigma_{l}$ is upper bounded: $\|\sigma_{l}\| \leq \prod_{q=1}^{l-1}B_{q} \|\sigma_{1}\|$, where $\sigma_{1}$ is the representation error of the input layer and $B_{q}$ is the Lipschiz-continuity parameter (see Assumptions 1 and 1.1). We empirically show that the term $\prod_{q=1}^{l-1}B_{q}$ is decaying and therefore the error $\|\sigma_{l}\|$ is likely to be vanishing. For illustration, Figs 9-10 (Appendix A) present the visualization of layer-by-layer error propagation. In these figures, we exhibit the representation errors of each layer and find that $\sigma_{l}$, i.e. the representation error on a layer $l$, is likely to decay as $l$ increases.
>
> However, if you feel that the empirical analysis is insufficient, we would like you to refer to Eq (16) and Appendix B.2 in paper [10] where an equivalent assumption is made. This paper first shows that the overall gradient error is bounded by a linear function of the last layer's gradient error, i.e. $\|\nabla f - \nabla f'\| \leq c_{1} \|\nabla f^{(N)} - \nabla f'^{(N)} \| + c_{2}$. ($c_{1}, c_{2}$ are constants and $N$ is the total layer number). Then the paper assumes $\|\nabla f'^{(N)} - \nabla f^{(N)} \|$ is bounded to therefore bound $\|\nabla f - \nabla f'\|$. This assumption is equivalent to the vanishing representation error justification in this paper. To prove this, first note $\|\nabla f^{(N)} - \nabla f'^{(N)}\| \leq L \|\sigma_{N-1}\| = L \prod_{q=1}^{N-1}B_{q}\|\sigma_{1}\|$ due to the $L$-smoothness of $f$. Then, if $\|\nabla f^{(N)} - \nabla f'^{(N)}\|$ is bounded, $\prod_{q=1}^{N-1}B_{q}$ must be shrinking, meaning that the representation error must be vanishing. Because by contradiction, if the errors are non-vanishing, $\prod_{q=1}^{N-1}B_{q}$ will explode as $N$ increases and $\|\nabla f^{(N)} - \nabla f'^{(N)}\|$ cannot be bounded.
>
> Therefore, we believe that vanishing representation error is a reasonable assumption as an equivalent assumption has already been made by existing works (we mention this work in lines 782-783, Appendix A).

---

> ### Author Response · Authors · 2024-11-20
> **Response to questions 1 and 2**
>
> Thank you for your valuable questions.
>
> __For Q1,__ as shown in Theorem 3 in the original TOA paper [11], the TOA error is proportional to 1/s − 1. Therefore, in terms of accuracy, a larger s always leads to better accuracy, and the optimal value of s is naturally one. However, in practice, the optimal value of s depends on how a client would like to balance accuracy between the actual memory footprint and communication cost, which is hardware-dependent.
>
> __For Q2__, in practice, changing the value of s has very little impact on the computation overhead. Because TOA is only applied to frozen layers, and varying s only affects the workload of forward propagation. However, forward propagation accounts for the overall computation overhead much less than forward propagation [16-18]. Therefore, it is unnecessary to vary s dynamically based on clients’ different hardware. Besides, the universal hyperparameter setting and grid search are common and unquestionable settings in many FL papers, such as [7, 8, 12, 13, 14, 15]

---

> ### Author Response · Authors · 2024-11-20
> **Response to questions 3-5**
>
> Thank you for your thoughtful questions.
>
> __For Q3__, as discussed in our previous response to Weakness 1, uniform client clustering is a common and widely used approach in federated learning experiments.
>
> __For Q4__, Regarding Section 3.5 and the convergence of FedOLF, we would like to clarify that FedOLF does not necessarily require any client to perform full-model training for convergence. This is supported by the findings (such as Figure 7) in the CKA paper [9]. It shows that for any two models, the CKA similarities between the very first few layers are usually high, even if one model is untrained and the other is fully trained. This means that, even if the first layer never gets trained, the output representation error will be quite limited compared with a trained model. Therefore, if none of the clients can do full-model training and the first few layers are forever frozen, FedOLF can still use these layers for forward propagation with limited training error. Moreover, the convergence analysis in Section 4 remains valid regardless of how many clients can do full training. Therefore, __FedOLF can still converge effectively without requiring clients to perform full-model training.__ The trade-off is that if fewer clients have full-model training capability, the accuracy will be somewhat lower due to increased gradient errors $D$ from
> frozen layers (see Section 4). However, the system will still converge to a solution, albeit with lower accuracy than if more clients were able to perform full training.
>
>
> __For Q5__, as stated in our response to Weakness 1, FedOLF is already evaluated in a scenario with a large proportion
> (80%) of resource-constrained clients and highly skewed class distributions (Dirichlet 0.1). For data-level assumptions, please refer to Assumption 3 (Eq. 9) which bounds the difference between local and global gradients.
> This is a common assumption that is pre-made by many existing works, such as [13, 14, 19, 20, 21].

---

> ### Author Response · Authors · 2024-11-20
> **Response to question 6**
>
> As shown in Figs. 11 and 13 in Appendix C, FedOLF has equivalent, even less training cost (FLOPs and energy) compared with other methods (the cost of training time is proportional to FLOPs). Moreover, due to its superior performance, FedOLF is able to achieve higher accuracy with the same amount of energy consumption (Fig. 4), which significantly improves both computation and communication efficiency.

---

> ### Author Response · Authors · 2024-11-20
> **Reference**
>
> 1. Samuel Horvath, Stefanos Laskaridis, Mario Almeida, Ilias Leontiadis, Stylianos Venieris, and Nicholas Lane. Fjord: Fair and accurate federated learning under heterogeneous targets with ordered dropout. In
> NeurIPS, 2021.
>
> 2. Enmao Diao, Jie Ding, and Vahid Tarokh. Heterofl: Computation and communication efficient federated learning for heterogeneous clients. In ICLR, 2021.
>
> 3. Minjae Kim, Sangyoon Yu, Suhyun Kim, and Soo-Mook Moon. DepthFL : Depthwise federated learning for heterogeneous clients. In ICLR, 2023.
>
> 4. Kilian Pfeiffer, Martin Rapp, Ramin Khalili, and Joerg Henkel. CocoFL: Communication- and computation-aware federated learning via partial NN freezing and quantization. Transactions on Machine Learning Research, 2023b.
>
> 5. Mi Luo, Fei Chen, Dapeng Hu, Yifan Zhang, Jian Liang, and Jiashi Feng. No fear of heterogeneity: Classifier calibration for federated learning with non-IID data. In NeurIPS, 2021
>
> 6. Weishan Zhang, Tao Zhou, Qinghua Lu, Yong Yuan, Amr Tolba, and Wael Said. Fedsl: A communication efficient federated learning with split layer aggregation. IEEE Internet of Things Journal, 2024.
>
> 7. Sebastian Caldas, Jakub Konecny, H. Brendan McMahan, and Ameet Talwalkar. Expanding the reach of federated learning by reducing client resource requirements. In NeurIPS Workshop on Federated Learning for Data Privacy and Confidentiality, 2018.
>
> 8. Kilian Pfeiffer, Ramin Khalili, and Joerg Henkel. Aggregating capacity in FL through successive layer training for computationally-constrained devices. In NeurIPS, 2023a.
>
> 9. Simon Kornblith, Mohammad Norouzi, Honglak Lee, and Geoffrey Hinton. Similarity of neural network representations revisited. In ICML, 2019.
>
> 10. Mirzasoleiman, Baharan, Jeff Bilmes, and Jure Leskovec. ”Coresets for data-efficient training of machine learning models.” In ICML, 2020.
>
> 11. Menachem Adelman, Kfir Levy, Ido Hakimi, and Mark Silberstein. Faster neural network training with approximate tensor operations. In NeurIPS, 2021.
>
> 12. Yue Tan, Guodong Long, Lu Liu, Tianyi Zhou, Qinghua Lu, Jing Jiang, and Chengqi Zhang. Fedproto: Federated prototype learning across heterogeneous clients. In AAAI, 2022.
>
> 13. Tian Li, Anit Kumar Sahu, Manzil Zaheer, Maziar Sanjabi, Ameet Talwalkar, and Virginia Smith. Federated optimization in heterogeneous networks. Proceedings of Machine learning and systems, 2020.
>
> 14. Ang Li, Jingwei Sun, Pengcheng Li, Yu Pu, Hai Li, and Yiran Chen. Hermes: An efficient federated learning framework for heterogeneous mobile clients. In Proceedings of ACM MobiCom, 2021.
>
> 15. R. Tamirisa et al., FedSelect: Personalized Federated Learning with Customized Selection of Parameters for Fine-Tuning, CVPR, 2024
>
> 16. Ziru Niu, Hai Dong, A.K. Qin. FedSPU: Personalized Federated Learning for Resource-constrained Devices with Stochastic Parameter Update, arXiv:2403.11464
>
> 17. Kaiming He and Jian Sun. 2015. Convolutional neural networks at constrained time cost. In CVPR, 2015.
>
> 18. Shen Li, Yanli Zhao, Rohan Varma, Omkar Salpekar, Pieter Noordhuis, Teng Li, Adam Paszke, Jeff Smith, Brian Vaughan, Pritam Damania, and Soumith Chintala. 2020. PyTorch Distributed: Experiences on Accelerating Data Parallel Training. Proc. VLDB Endow. 13, 2020.
>
> 19. Sai Praneeth Karimireddy, Satyen Kale, Mehryar Mohri, Sashank Reddi, Sebastian Stich, and Ananda
> Theertha Suresh. SCAFFOLD: Stochastic controlled averaging for federated learning. In ICML, 2020.
>
> 20. Zhida Jiang, Yang Xu, Hongli Xu, Zhiyuan Wang, Jianchun Liu, Qian Chen, and Chunming Qiao. Computation and communication efficient federated learning with adaptive model pruning. IEEE Transactions on Mobile Computing, 2023.
>
> 21. Canh T. Dinh, Nguyen H. Tran, and Tuan Dung Nguyen. Personalized federated learning with moreau envelopes. In NeurIPS, 2020.
>
> 22. Tan, Jiahao, et al. "pFedSim: Similarity-Aware Model Aggregation Towards Personalized Federated Learning." arXiv preprint arXiv:2305.15706.

---

> ### Author Response · Authors · 2024-11-25
>
> Dear reviewer,
>
> Thanks for your response.
>
> First, __we would like to justify that the statement__ “This statement clearly states the limitations of the proposed technique” __may not be appropriate__. Accuracy decay along with the reduction of full-model clients is a common property of SOTA works and __is not unique__ in our approach. This has been verified by extensive experiment results in the baselines' papers, such as: Fig.3 in Feddrop [7], Fig. 7 in FjORD [1], Table 5 in DepthFL [3], Table 1 in HeteroFL [2], Table 2 in CoCoFL [4] and Table 3 in SLT [8].
>
> Second, __to answer the questions__ “Also, can this technique be used when most of the devices are significantly memory constrained. If yes, what are the limits? Also, how does the proposed technique compare with the state-of-the-art techniques under such scenarios." __and__ "The non-uniform clustering is just to identify the limitations of the proposed technique in the context where only a very limited number of devices (e.g., just one or none) have full model training capability":
>
>
> __we’ve reduced the number of full-model clients to one (they are replaced by clients who can only train 20\% of the entire model) and evaluated FedOLF on CIFAR-100 / ResNet-20 and got 37.62% accuracy, which is only 0.23% lower than the uniform case__ (see Table 1), which shows the robustness of FedOLF to extreme non-uniform cases where most clients are resource-constrained. Due to a limited response time, we are unable to provide the accuracy of the baselines in the non-uniform case. However, we can see that FedOLF in non-uniform case still outperforms the baselines in uniform case (Table 1):
>
> | Feddrop | FjORD | HeteroFL | DepthFL | CoCoFL | SLT    | FedOLF(non-uniform) |
> |---------|-------|----------|---------|--------|--------|---------------------|
> | 17.02%  | 12.7% | 12.32%   | 24.87%  | 22.16% | 25.04% | 37.62%              |
>
> It has been well-studied that the baselines’ performance always decrease when it goes from uniform case to non-uniform case (see examples above). Therefore, the SOTA’ accuracy in non-uniform case will definitely be lower than the uniform case, and hence lower than FedOLF in non-uniform case. To our best knowledge, we haven’t seen any possible limits in the non-uniform scenario.
>
> Lastly, the question “If state-of-the-art techniques outperform the proposed under such scenarios, what scenarios are ideal for the proposed technique?” has also been addressed as SOTA methods will not outperform FedOLF.
>
> Regards

---

> ### Author Response · Authors · 2024-11-28
> **Response to reviewer's questions**
>
> We respectfully address the points raised in your latest comments, ensuring clarity and rigor in our explanations.
>
> ### 1. Harsh Memory Constraints
>
> Your statement that “the authors have not tested the concept under harsh memory constraints” appears to overlook our initial response, where we explicitly outlined an experimental setup involving __80% memory-constrained devices__—a scenario widely recognized in the literature as a harsh memory-constrained setting [1-3]. Additionally, we conducted further experiments as you requested, reducing the number of full-model clients to just one, and our results remained strong. If this does not satisfy the standard for "harsh memory constraints," we kindly request you to clarify __what would constitute an acceptable threshold__, along with __relevant citations__.
>
> ### 2. Computation and Communication Constraints
>
> Regarding the claim that "memory being the most important constraint" was supported by only a few references, we again encourage you to provide the additional references that demonstrate the equal importance of computational, memory, and communication constraints in FL). Although FedOLF focuses on memory constraints as they are the most pertinent factors to determine whether edge devices can do full-model training, the computation and communication constraints are NOT overlooked in this paper. First, OLF simultaneously mitigates computation overhead by minimizing the length of backward propagation path (Fig. 1). Second, we propose TOA to further reduce computation and communication overheads. Thirdly, __your statement that we completely ignore the computation and communication constraints is contradictory to your initial review, wherein it is clearly written “Applying TOA to frozen layers preserves model accuracy while reducing the communication cost”__.
>
> ### 3. Diverse Non-iid Data distributions
>
> As stated in your initial review, the __goal__ of testing FedOLF with diverse non-IID data distribution __is to validate the claim that "low-level layers universally obtain higher CKA similarity."__ In our last response, we showed that this concern is misplaced, as the CKA similarity property has already been substantiated in multiple prior works ([5,6,9]). Therefore, CKA similarity is a verified finding, not an untested assumption. In this case, no extra tests are needed for any verification, and reiterating the requirement of diverse non-iid evaluation seems to be unreasonable.
>
> Furthermore, all datasets we employed (EMNIST, CIFAR10, CIFAR100, and CINIC10) are NOT “additional”, as the CKA theory is proven to work well on these datasets [5,9,22].
>
> ### 4. Comparison to CoCoFL
> You mentioned that layer freezing were proposed in CoCoFL ([4]). However, the main contribution of our paper lies in ordered layer freezing rather than layer freezing only. We have introduced many aspects wherein our method (ordered freezing) differs from CoCoFL, including:
> * Backward gradient path minimization (Figure 1)
> * Low-level layer sharing (Section 3.5)
> * Limited training error (Section 3.6)
> * Tensor Operation Approximation (TOA) (Section 3.3)
>
> These innovations result in significant performance improvements, as demonstrated in our evaluations. Thus, FedOLF represents a distinct and superior approach, not merely an extension of CoCoFL. Besides, __“layer freezing and quantization have been proposed in COCOFL” is a new concern that you didn’t raise in the original review, which should not be proposed in the discussion phase.__ Introducing new critiques during the discussion phase risks undermining the review process's integrity.
>
> ### 5. Handling Computational Constraints
> Regarding devices with limited computation power, our method (and SOTA) employs a __synchronous FL protocol__—a standard practice in the field. Under this protocol, the server waits for all clients to complete training. We have demonstrated that our method does not increase overall training time compared to baselines (Figure 11, Appendix C). While latency is a limitation of synchronous FL, it is not unique to our method and should not be grounds for rejection.
>
> ### 6. Inconsistency Between Comments
>
> By saying “all computational, memory, and communication constraints have to be considered for the system design. See [4]”, you admit that [4]’s setting (layer freezing, uniform clustering) presents an eligible system design in overall aspects. This is contradictory to your another statement “such constraints have been completely ignored by the authors”, as our work follows the setting of [4].
>
> ### 7. Review Integrity
> Lastly, __we sincerely would like you to explain why you didn’t propose these questions in your last response.__ As a matter of integrity, reviewers are expected to raise all questions once, allowing authors sufficient time to address them comprehensively.

---

### Official Review · Reviewer_sXem · 2024-11-04

**Soundness:** 2
**Presentation:** 3
**Contribution:** 2
**Rating:** 6
**Confidence:** 3

**Summary:**

The edge applications leveraging FL are highly constrained by resources, in terms of computation and communication. Existing efficient FL solutions either compromise accuracy or ignore memory usage. They introduce the ordered-layer-freezing and Tensor Operation Approximation for reducing memory footprint & communication overhead.

**Strengths:**

Tensor Operation Approximation is used to reduce the communication overhead of FL.
Various test results across small datasets are demonstrated.

**Weaknesses:**

layerfeezing technique is not new. Novelty seems to be marginal

**Questions:**

No.

---

> ### Author Response · Authors · 2024-11-20
> **Author Response**
>
> Dear reviewer,
>
> Thank you for reviewing the paper.
>
> We appreciate the reviewer’s observation regarding the prevalence of layer freezing in machine learning. While it is true that layer freezing has been explored extensively in centralized machine learning, our contribution specifically addresses its application in the FL paradigm, which presents unique challenges and dynamics not encountered in centralized settings. To the best of our knowledge, our work is the first to propose Ordered Layer Freezing (OLF) in FL, emphasizing the systematic ordering of layer updates based on task-specific importance—a key differentiator from existing works.
>
> We acknowledge that high-level layers often contribute more to task-specific learning than low-level layers, as noted. However, our approach goes beyond this established understanding by (1) providing a principled strategy for ordering updates in FL to optimize both resource efficiency and learning performance, and (2) demonstrating its benefits with rigorous quantitative analyses, including memory footprint reduction (Fig. 1), generalized feature sharing across low-level layers (Section 3.5), and detailed convergence behavior (Section 4). These analyses validate the effectiveness and novelty of OLF in the FL context, where communication constraints and heterogeneous client data necessitate innovative adaptation of layer freezing.
>
> Regards

---

### Meta-Review · Area_Chair_NTgF · 2024-12-14

**Metareview:**

This study proposes to replace the random layer freezing with the ordered layer freezing and adopts a tensor operation approximation technique to reduce memory footprint and better keep accuracy for the problem of federated learning in IoT systems. The proposed method has a clear motivation and its effectiveness in terms of higher accuracy and energy efficiency is demonstrated by extensive experiments. However, a major issue as raised unanimously by all the three reviewers is the technical novelty. Both layer freezing and tensor operation approximation are not new techniques. The AC looks through the paper and all the discussions, and agrees that the study needs improvement regarding its technical contribution.

**Additional Comments On Reviewer Discussion:**

All three reviewers unanimously raise the concern about the method novelty. During rebuttal, the authors explain and highlight that they apply the method to the new problem of FL. But the reviewers maintain their opinions. The AC also agrees that the issue is not fully addressed.

---

### Decision · Program_Chairs · 2025-01-22

Reject